# Ocean-atmosphere interactions modulate irrigation's climate impacts

**Nir Y Krakauer**[1]**, Michael J Puma**[2,3,4]**, Benjamin I Cook**[3]**, Pierre Gentine**[5]**, and Larissa Nazarenko**[2,3]

[1]Department of Civil Engineering and NOAA-CREST, The City College of New York, New York, NY, 10031, USA
[2]Center for Climate Systems Research, Columbia University, New York, NY 10025, USA
[3]NASA Goddard Institute for Space Studies, New York, NY 10025, USA
[4]Center for Climate and Life, Columbia University, Palisades, NY 10964, USA
[5]Earth and Environmental Engineering, School of Engineering and Applied Science, Columbia University, New York, New York, USA

*Correspondence to:* NY Krakauer (mail@nirkrakauer.net)

**Abstract.** Numerous studies have focused on the local and regional climate effects of irrigated agriculture and other land cover and land use change (LCLUC) phenomena, but there are few studies on the role of ocean-atmosphere interaction in modulating irrigation climate impacts. Here, we compare simulations with and without interactive sea surface temperatures of the equilibrium effect on climate of contemporary (year 2000) irrigation geographic extent and intensity. We find that ocean-atmosphere interaction does impact the magnitude of global-mean and spatially varying climate impacts, greatly increasing their global reach. Local climate effects in the irrigated regions remain broadly similar, while non-local effects, particularly over the oceans, tend to be larger. The interaction amplifies irrigation-driven standing wave patterns in the tropics and midlatitudes in our simulations, approximately doubling the global mean amplitude of surface temperature changes due to irrigation. The fractions of global area experiencing significant annual-mean surface air temperature and precipitation change also approximately double with ocean-atmosphere interaction. Subject to confirmation with other models, these findings imply that LCLUC is an important contributor to climate change even in remote areas such as the Southern Ocean, and that attribution studies should include interactive oceans and need to consider LCLUC, including irrigation, as a truly global forcing that affects climate and the water cycle over ocean as well as land areas.

## 1 Introduction

Anthropogenic land cover and land use change (LCLUC) affects climate by modifying water, sensible heat, and radiation fluxes at the land surface (Chase et al., 2000; Gordon et al., 2005; Brovkin et al., 2006; Findell et al., 2007; Krakauer et al., 2010; Mahmood et al., 2014). One important mode of LCLUC has been the dramatic expansion in irrigated agriculture over the past century. Resultant local climate changes, notably growing-season daytime cooling resulting primarily from increased evapotranspiration, have been diagnosed from observations (Bonfils and Lobell, 2007; Lobell and Bonfils, 2008; Misra et al., 2012). Remote (non-local) impacts of irrigation are less well constrained. Global climate models (GCMs) can be run with and without an irrigation scheme to assess local climate effects as well as remote impacts (such as downwind enhancement of precipitation), which would be difficult to deduce with confidence from observations alone because the propagation mechanisms may not be easily observable and because trends in observations are often dominated by the effects of other climate forcings (Lo et al., 2013; Alter et al., 2015; de Vrese et al., 2016).

Many GCM studies of irrigation's climate impacts have been conducted with prescribed sea surface temperatures (SSTs) (Boucher et al., 2004; Puma and Cook, 2010; Lo and Famiglietti, 2013; de Vrese et al., 2016), while several did include ocean-atmosphere interaction (Lobell et al., 2006; Cook et al., 2011, 2015). Various studies have highlighted the importance of interactive atmosphere-ocean coupling for accurately reproducing various phenomena in GCMs. These

include Indian monsoon rainfall (Kumar et al., 2005; Wu and Kirtman, 2004; Shukla et al., 2014) and the relationship between sea level pressure and SST trends (Copsey et al., 2006; Meng et al., 2012). Further, the oceans may be important for modulating responses from LCLUC forcings. For example, studies of afforestation and deforestation at high Northern latitudes show that responses to these LCLUC forcings are amplified in simulations that included interactive SSTs, as fixed SSTs damp positive feedbacks that involve changes in ocean temperature and sea ice cover (Bonan et al., 1992; Swann et al., 2010). To date, however, no studies have explicitly compared the effect of interactive versus prescribed SSTs on model responses to realistic irrigation forcing.

In this study, therefore, we examine the possible role of atmosphere-ocean interaction in modulating the impact of irrigation on climate. We conduct GCM simulations of steady-state climate with and without present-day irrigation extents and with either prescribed SSTs or a thermodynamic slab ocean model. Broadly, consistent with the previous work on LCLUC, we expected the interactive SST configuration to allow more of the Earth system to respond to the irrigation forcing. By contrast, fixed SSTs would tend to act as a stabilizing influence that limits the degree to which forcings such as irrigation can impact climate.

## 2   Methods

### 2.1   Model runs

We analyze different model experiments to investigate irrigation forcing of climate, all using the GCM ModelE2 ($2°$ latitude $\times$ $2.5°$ longitude resolution), the latest version of the GISS atmosphere general circulation model, with 40 vertical layers in the atmosphere and updated physics (Schmidt et al., 2014; Miller et al., 2014). Irrigation water is added to the vegetated fraction of the grid cell at the top of the soil column, beneath the vegetation canopy. Irrigation rates are nominally for the year 2000, taken from a global gridded reconstruction (Wisser et al., 2010) (Figure 1). This reconstruction estimates irrigation demand based on combining maps of irrigated areas and crop types with crop-specific evapotranspiration scale factors, with a special allowance for maintaining a constant flood depth in paddy rice areas (Wisser et al., 2010). Water for irrigation is initially withdrawn from rivers and lakes in the same grid cell. If irrigation demand is not satisfied by these surface sources, water is added under the assumption that it is taken from groundwater sources that are not represented in the model (i.e., 'fossil' groundwater). The irrigation rate is kept constant over the course of the day and applied for every sub-daily time step. Irrigation water will either infiltrate the soil column or run off to the streams in the grid cell. The total amount of irrigation water averaged 0.019 mm per day (6.8 mm per year) globally (3500 km³ per year total), with a mean of 0.46 mm per day (168 mm per year) over irrigated land grid cells (defined as those

for which the average irrigation amount was at least 0.1 mm per day). Globally some 42% of the applied irrigation water (1500 km³ per year) was modeled as coming from groundwater, a proportion similar to that found in an independent modeling study of the fraction of consumptive irrigation water use deriving from groundwater (Siebert et al., 2010). Additional details and discussion of the irrigation scheme are in Puma and Cook (2010) and Cook et al. (2011). As opposed to 'Irrigation' (irrig) runs, in 'Control' (ctrl) runs no irrigation water is applied. Forcings such as greenhouse gas concentrations were the same for all the experiments, and based on values from around the year 2000 (Cook et al., 2011).

Irrigation and Control simulations were carried out with two different ocean configurations. The first involves forcing the atmosphere model with prescribed, annually repeating monthly sea surface temperatures (SSTs) and sea ice. The SSTs and sea ice are based on average 1996 to 2004 data from the Hadley Center analysis (Rayner et al., 2003). We refer to this as the atmosphere-only, fixed-SST, or A configuration. In the second configuration (referred to as 'q-flux' mode, interactive-SST, interactive-(surface)-ocean, or O configuration), the ocean is represented as a 65-m deep mixed layer, with a prescribed distributed heat source to represent the effects of horizontal and vertical ocean mixing and advection.

The four simulations – irrig-A, ctrl-A, irrig-O, ctrl-O – were run 60 years each. The q-flux mode takes approximately 10 years to reach equilibrium under constant forcings, so we analyzed only the last 50 years of each simulation, which represent approximately steady-state conditions that show internal system variability under the different model configurations (i.e., with fixed SST or q-flux ocean, and with or without irrigation). Figure 2 illustrates the approach to equilibrium of the simulations. The A runs stayed at essentially the same temperature (with internal year-to-year variability) from the first year, but had 0.4 W per m² more radiation entering Earth than leaving. This was because the observation-based fixed SST was cooler than needed for radiative equilibrium with the imposed greenhouse gas concentration (Hansen et al., 2005), although surface temperature and other climate variables did remain at steady state within the fixed-SST model configuration. In the O runs, the radiative imbalance largely resolved itself within a few years by SSTs and surface air temperatures warming around 0.3 K, and a difference of 0.1 K between the irrig and ctrl runs in the equilibrium mean temperature was evident (Figure 2).

### 2.2   Analysis of differences between runs

For climate variables of interest, we considered irrig−ctrl differences in the monthly fields for both the A and O configurations. The irrig-ctrl difference field for the A set of experiments is referred to as $\Delta_A$, and the irrig-ctrl difference field for the O set of experiments is referred to as $\Delta_O$. The

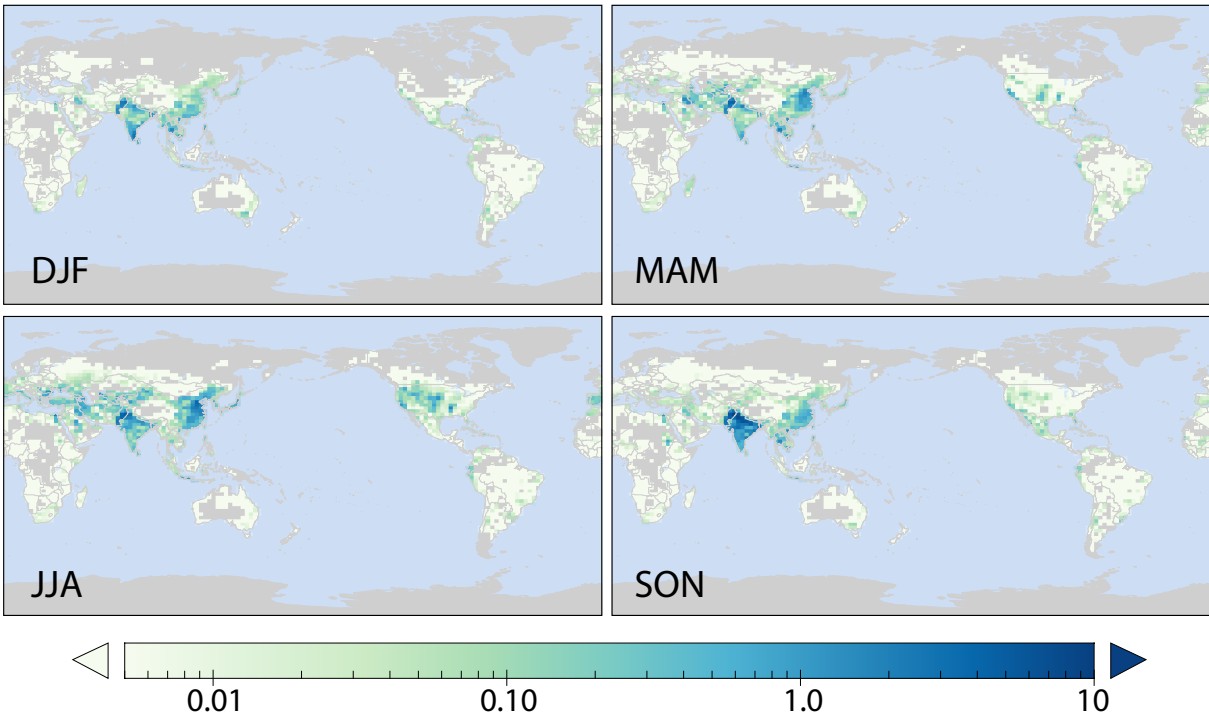

**Figure 1.** Applied irrigation by season (mm per day). The scale is logarithmic. Land with no irrigation applied is shown in gray.

impact of interactive SSTs on the equilibrium irrig-ctrl difference was then obtained as $\Delta\Delta \equiv \Delta_O - \Delta_A$.

The significance of differences $\Delta_A, \Delta_O, \Delta\Delta$, either at individual grid points or spatially averaged, was estimated using a Student's t-test on their time series over the 50-year analysis period, with the degrees of freedom adjusted based on the lag-1 autocorrelation of the time series. This adjustment is based on the notion of effective sample size in time series analysis, taking as the null hypothesis that the difference time series is red noise with zero mean (Jones, 1975; Bretherton et al., 1999).

As metrics of overall irrigation and ocean configuration impacts, we looked at the mean and the root mean square (rms) of $\Delta_A, \Delta_O, \Delta\Delta$ aggregated globally over irrigated areas (which we defined as grid cells and months where the applied irrigation was over 0.1 mm day$^{-1}$); non-irrigated land areas; and ocean areas. We considered annual means of these quantities as well as seasonal means. For seasonal means, aggregation was performed only over the zone of the Northern Hemisphere where the vast majority of the irrigation takes place (8°-46°N, 92% of global irrigation), to preserve consistent seasonality.

We focus on model climate variables that quantify directly conditions and moisture status at Earth's surface (surface air temperature, SST [only over ocean], precipitation, soil moisture [only over land], cloud fraction); terms in the surface energy balance that are affected by irrigation (latent and sensible heat fluxes); and circulation-related quantities (sea-level pressure, geopotential height, and meridional jet stream velocity fields) that can provide insight into how irrigation effects on surface energy and water balance could propagate to impact climate in distant regions. (The jet stream velocity is defined to be that at a pressure of around 250 mb, in the upper troposphere.)

## 3 Results

### 3.1 Impact of interactive SST on spatial-mean irrigation responses

The irrigation-induced surface air cooling, though still concentrated over irrigated land areas, spread over ocean areas in the interactive-SST simulation, consistent with our expectation that fixing SST would tend to stabilize air temperatures over the ocean. Global-mean above-ocean surface air temperatures and sea surface temperatures both decreased 0.08 K (Table 1). In the fixed-SST irrigation simulation, precipitation slightly decreased over the irrigated areas and increased elsewhere. Compared to the fixed-SST irrigation simulation, the cooling over the oceans slightly reduced evaporation and precipitation in the interactive-SST simulation. Interactive SST did not significantly modify the global mean enhancements in soil moisture and cloudiness due to irrigation (Table 1). Irrigation-induced latent and sensible surface heat fluxes were both slightly diminished in the interactive-SST simu-

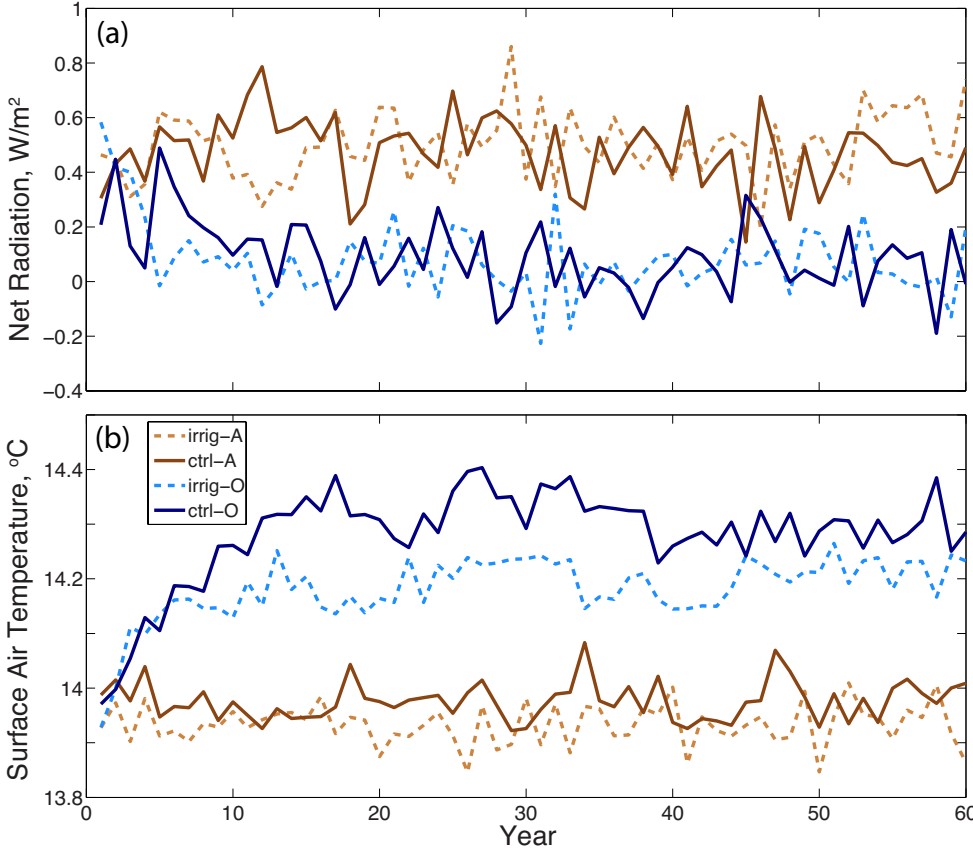

**Figure 2.** Global and annual mean (a) top-of-atmosphere net radiation, (b) surface air temperature over the four simulations.

lation, consistent with the cooler surface temperatures and reduced precipitation (Table 1). The mean atmospheric pressure responded inversely to the temperature change, with higher pressure in the cooled irrigated areas (consistent with the reduced precipitation there). The mean 300-mb height decreased significantly more in the interactive-SST simulation even in the irrigated areas, showing that, compared to fixed SST, interactive SST spreads the cooling due to irrigation throughout the atmospheric column (Table 1). The meridional jet stream velocity was slightly higher in the runs with irrigation, although the effect of interactive SST ($\Delta\Delta$) was only significant over irrigated areas (Table 1).

Table 2 shows changes by season (averaged over 8°-46°N) for surface air temperature. Over land, the cooling is greatest in the summer and fall, when the largest amount of irrigation water is applied, and the mean amount is not significantly affected by whether SST is interactive. Over the ocean, cooling is more uniform across seasons, and is much greater in the interactive SST simulation (Table 2).

## 3.2 Impact of interactive SST on spatial variability of irrigation responses

The global or Northern Hemisphere mean impacts just shown conceal much spatial variability in the response to irrigation. The rms of the spatial field of irrigation response for the same climate variables (Table 3) shows that interactive SST tends to increase this spatial variability over the ocean and non-irrigated land, even for variables such as over-ocean cloud cover and jet stream velocity for which the mean response is not significantly affected, implying that interactive SST on the whole enhances non-local irrigation impacts on climate. One exception is that interactive SST decreases the spatial variability in latent and sensible heat irrigation responses over the ocean (Table 3), presumably because the interactive SST adjusts to changes in air temperature in a way that reduces the equilibrium change in surface fluxes.

We show illustrative maps of the seasonal-mean irrigation response with and without interactive SST ($\Delta_A, \Delta_O$). Under fixed SST, irrigation-induced changes in surface air temperature (Figure 3) are primarily local to major irrigation regions such as India, China, and the United States (USA), and effects in the ocean tend to be small, except in the North

**Table 1.** Mean impact of irrigation on climate quantities with and without interactive sea surface temperatures.

| | Irrigated land | | | | Non-irrigated land | | | | Ocean | | | |
| --- | --- | --- | --- | --- | --- | --- | --- | --- | --- | --- | --- | --- |
| | Mean | $\Delta_A$ | $\Delta_O$ | $\Delta\Delta$ | Mean | $\Delta_A$ | $\Delta_O$ | $\Delta\Delta$ | Mean | $\Delta_A$ | $\Delta_O$ | $\Delta\Delta$ |
| Surface air temperature (°C) | 18.764 | -0.665*** | -0.697*** | -0.032 | 7.760 | -0.078** | -0.123*** | -0.045 | 16.008 | -0.003 | -0.084*** | -0.081*** |
| Sea surface temperature (°C) | | | | | | | | | 21.676 | 0 | -0.076*** | -0.076*** |
| Precipitation (mm d$^{-1}$) | 3.085 | -0.032 | -0.082*** | -0.050 | 1.935 | 0.032*** | 0.036*** | 0.005 | 3.371 | 0.011*** | 0.005** | -0.006* |
| Soil moisture (mm) | 462.7 | 60.5*** | 56.6*** | -3.9 | 416.0 | 7.2*** | 9.7*** | 2.5 | | | | |
| Cloud cover (%) | 50.27 | 1.28*** | 0.96*** | -0.32 | 59.13 | 0.43*** | 0.33*** | -0.10 | 63.37 | 0.07** | 0.09* | 0.02 |
| Latent heat (W m$^{-2}$) | 55.56 | 9.08*** | 8.38*** | -0.70*** | 39.07 | 0.63*** | 0.51*** | -0.12 | 104.93 | 0.03 | -0.08** | -0.11* |
| Sensible heat (W m$^{-2}$) | -47.99 | 6.03*** | 5.49*** | -0.55** | -38.35 | 0.48*** | 0.44*** | -0.04 | -18.80 | 0.05*** | -0.03** | -0.08*** |
| Sea-level pressure (mb) | 1010.36 | 0.47*** | 0.39*** | -0.08 | 1010.16 | 0.11 | 0.02** | -0.09 | 1009.89 | -0.05*** | -0.01 | -0.04** |
| 300-mb height (m) | 9459.55 | -1.90** | -4.58*** | -2.67** | 9207.85 | -0.11 | -3.11*** | -3.00*** | 9316.66 | -0.05 | -1.76*** | -1.71*** |
| Meridional jet (m s$^{-1}$) | 20.36 | +0.29*** | +0.49*** | +0.19* | 14.13 | +0.06 | +0.06 | +0.01 | 15.65 | +0.14*** | +0.20** | +0.05 |

Means are for the ctrl-A (no irrigation, fixed SST) simulation. Significance level (two-tailed) of differences: *0.05, **0.01, ***0.001.

**Table 2.** Mean impact of irrigation on seasonal surface air temperature (°C, averaged over 8°-46°N) with and without interactive sea surface temperatures.

| | Irrigated land | | | | Non-irrigated land | | | | Ocean | | | |
| --- | --- | --- | --- | --- | --- | --- | --- | --- | --- | --- | --- | --- |
| | Mean | $\Delta_A$ | $\Delta_O$ | $\Delta\Delta$ | Mean | $\Delta_A$ | $\Delta_O$ | $\Delta\Delta$ | Mean | $\Delta_A$ | $\Delta_O$ | $\Delta\Delta$ |
| Winter (DJF) | 10.407 | -0.607*** | -0.633*** | -0.026 | 10.302 | -0.036 | -0.130 | -0.094 | 19.282 | +0.000 | -0.188*** | -0.188*** |
| Spring (MMA) | 19.414 | -0.622*** | -0.663*** | -0.041 | 18.530 | -0.118* | -0.165** | -0.047 | 20.522 | -0.021*** | -0.170*** | -0.149*** |
| Summer (JJA) | 26.242 | -0.674*** | -0.726*** | -0.052 | 26.333 | -0.329*** | -0.396*** | -0.066 | 24.432 | -0.030*** | -0.178*** | -0.147*** |
| Fall (SON) | 19.474 | -0.885*** | -0.891*** | -0.006 | 19.580 | -0.225*** | -0.307*** | -0.082 | 23.487 | -0.020*** | -0.163*** | -0.143*** |

Means are for the ctrl-A (no irrigation, fixed SST) simulation. Significance level (two-tailed) of differences: *0.05, **0.01, ***0.001.

**Table 3.** Root mean square impact of irrigation on time-mean climate quantities with and without interactive sea surface temperatures.

| | Irrigated land | | | Non-irrigated land | | | Ocean | | |
| --- | --- | --- | --- | --- | --- | --- | --- | --- | --- |
| | $\Delta_A$ | $\Delta_O$ | | $\Delta_A$ | $\Delta_O$ | | $\Delta_A$ | $\Delta_O$ | |
| Surface air temperature (°C) | 1.289 | 1.267 | - | 0.989 | 1.047 | * | 0.374 | 0.769 | *** |
| Sea surface temperature (°C) | | | | | | | 0 | 0.549 | *** |
| Precipitation (mm d$^{-1}$) | 0.985 | 0.977 | - | 0.547 | 0.590 | ** | 0.847 | 0.916 | *** |
| Soil moisture (mm) | 120.6 | 119.0 | - | 52.9 | 59.0 | *** | | | |
| Cloud cover (%) | 4.64 | 4.72 | - | 4.01 | 4.10 | - | 3.07 | 3.46 | *** |
| Latent heat (W m$^{-2}$) | 17.31 | 17.01 | * | 5.90 | 6.19 | * | 7.96 | 7.55 | *** |
| Sensible heat (W m$^{-2}$) | 12.14 | 11.94 | - | 6.22 | 6.58 | ** | 3.55 | 3.40 | * |
| Sea-level pressure (mb) | 1.01 | 0.97 | - | 1.66 | 1.47 | - | 1.45 | 1.40 | - |
| 300-mb height (m) | 17.31 | 17.68 | - | 25.72 | 24.93 | - | 22.27 | 23.84 | * |
| Meridional jet (m s$^{-1}$) | 2.24 | 2.31 | - | 2.14 | 2.21 | - | 2.28 | 2.57 | *** |

Significance level (two-tailed) of differences due to interactive sea surface temperature: -not significant ($p > 0.05$), *0.05, **0.01, ***0.001.

Pacific. Under interactive SST, irrigation-induced regional changes tend to have larger amplitude (∼0.8 compared to ∼0.4 K; Table 3) and are also found in the tropical and southern oceans. Under fixed SST in boreal winter, the middle and high northern latitudes show a stationary wave pattern of alternating warm and cool anomalies due to irrigation (which during that season is concentrated in the Indian subcontinent). Under interactive SST, these boreal winter anomalies shift locations somewhat (for example, the cooling centered in the eastern USA under fixed SST is attenuated) and persist to a greater extent during the other seasons, and analogous wave patterns in the Southern Ocean and Antarctica are considerably stronger than under fixed SST. Under interactive SST, surface air temperature anomalies outside irrigated areas tend to be closely associated with SST anomalies of the same sign (Figure 4), which provide a mechanism for the surface air temperature anomalies to persist across seasons and supports the role of air-sea interactions in driving the divergence in irrigation responses between the fixed-SST and interactive-SST simulations..

Under fixed SST, a reduction in 300-mb height (corresponding to cooling of the atmospheric column; Figure 5) is seen primarily around irrigation regions in the northern midlatitudes, while under interactive SST the reduction in the northern midlatitudes is more zonally uniform, and there is also a stronger stationary wave pattern in the Southern Hemisphere roughly corresponding to locations of surface air temperature changes there. Particularly in boreal winter, the stationary wave pattern seen for surface air temperature is also found in the upper atmosphere, with phases shifted between the interactive and fixed SST simulations (Figure 5). The meridional jet stream velocity ($u_{\mathrm{jet}}$) changes correspondingly, consistent with geostrophic adjustment of the atmospheric circulation: $u_{\mathrm{jet}}$ tended to increase on the north side, and decrease on the south side, of areas where 300-mb geopotential height rose, and vice versa where geopotential height dropped (Figure 6). The changes seen in 300-mb height are of order 20 m or ∼0.2%, while the changes seen in meridional jet stream velocity are of order 2 m s$^{-1}$ or ∼10% (cf. Tables 1 and 3).

Precipitation impacts (Figure 7) are strongest over the tropics and subtropics and appear to reflect, for example, a northward shift due to irrigation in the intertropical convergence zone (ITCZ) in and south of India in boreal winter and a southward shift in boreal summer, with the zonal-mean effect of irrigation under interactive SST being to decrease tropical precipitation north of the Equator and increase it south of the Equator. The summer monsoon precipitation over India is reduced under both fixed and interactive SST, but with interactive SST impacts of irrigation on summer precipitation appear to also be more widespread across southeast Asia (Figure 7). Latent heat impacts (Figure 8) reflect both increased evapotranspiration where there is irrigation and the impacts of nonlocal changes in temperature and precipitation, e.g. less evaporation over western Australia in Austral

summer associated with reduced precipitation there due to irrigation under interactive SSTs.

One measure of the increased modeled global impacts of irrigation under interactive SST is provided by the fraction of the global area with significant ($p < 0.05$) change in the annual mean of each climate variable due to irrigation with fixed SST ($\Delta_A$) versus interactive SST ($\Delta_O$). This area fraction increases substantially with interactive SST for most of the variables discussed here, for example more than doubling (21% to 46%) for surface air temperature and almost doubling (15% to 27%) for precipitation.

## 4  Discussion

The present work suggests that an interactive-SST (q-flux) global model configuration, compared to one with fixed SSTs, results in similar mean local climate effects in the irrigated regions, but generally larger non-local effects, particularly over the oceans. In response to the application of realistic present-day irrigation amounts, the q-flux configuration generates stationary wave patterns across a range of latitudes in climate variables such as surface air temperature, SST, and geopotential height. These wave patterns have fairly large amplitudes (e.g. up to ∼1 K in SST, similar to the magnitude of warming from anthropogenic greenhouse gas emissions over the past century). The stationary waves generated are qualitatively similar to those previously studied as occurring in response to zonal asymmetries (Held et al., 2002; Shaman and Tziperman, 2005). An atmosphere-only GCM study (Koster et al., 2014) identified phase locking and amplification of a planetary wave as a potential mechanism for nonlocal climate impacts of soil moisture changes (such as those imposed by irrigation) in boreal spring and summer over the continental USA, but did not attempt to assess to what extent such feedbacks are likely to be affected by air-sea interactions. In our simulations, these patterns are less pronounced when SST is fixed, implying that air-sea interaction is essential to their propagation and maintenance, and are seen even at locations such as the Southern Ocean that are far from most of the irrigated areas.

While comparison with such past studies suggests that the occurrence of stationary waves amplified by air-sea interaction in response to irrigation is likely robust, their location and magnitude may be sensitive to, for example, aspects of our atmosphere model parametrization, background climate and ocean fluxes, and details of how the irrigation is applied. Systematic multi-model intercomparisons of responses to irrigation and other LCLUC forcings could aid in understanding these sensitivities, illuminate the physical mechanisms at play, and identify suitable targets for testing modeled LCLUC-induced non-local climate change against observations.

The impacts of air-sea interaction on irrigation effects on tropical and monsoon precipitation are qualitatively consis-

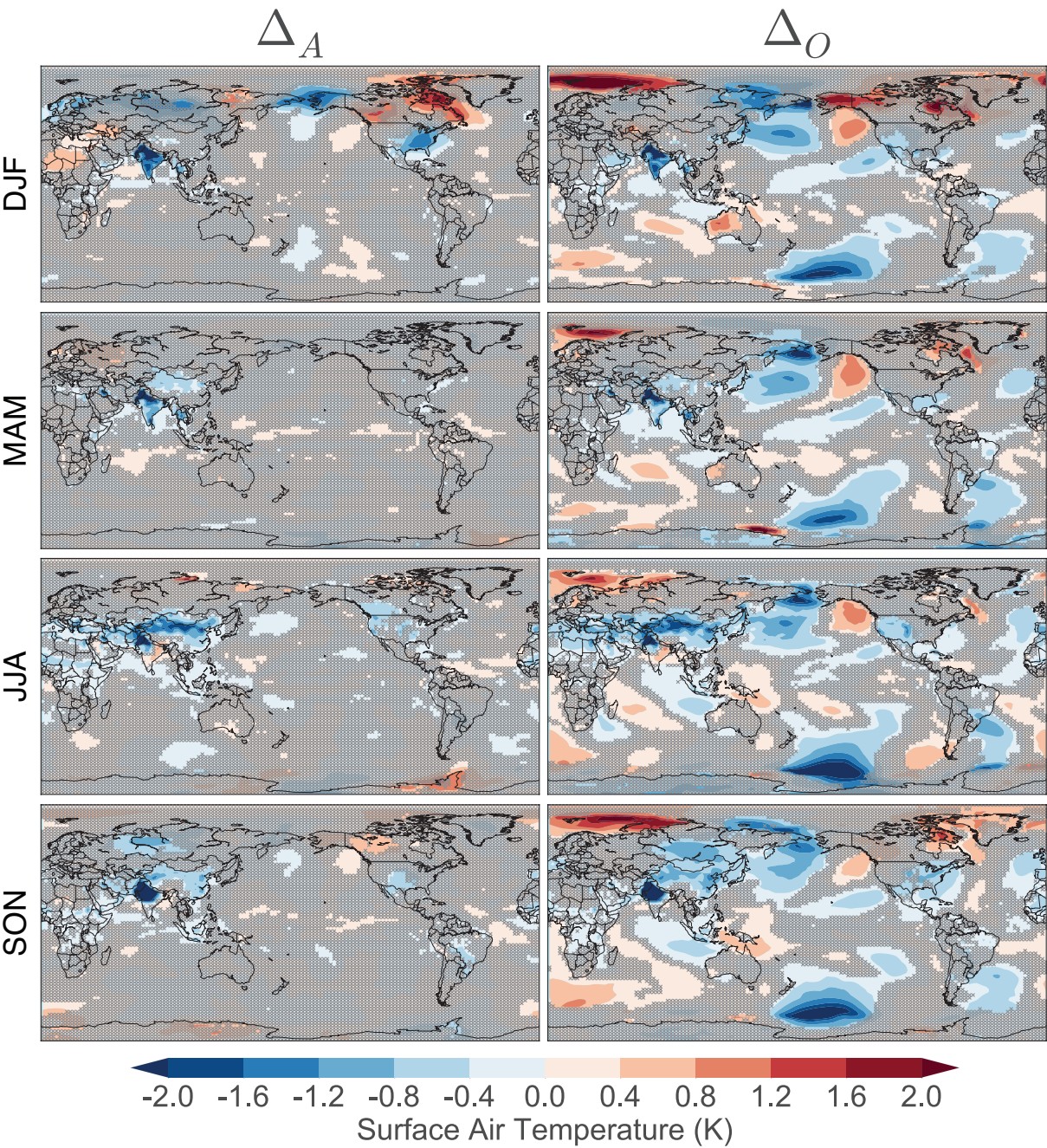

**Figure 3.** Difference in surface air temperature (K) by season due to irrigation with fixed SST ($\Delta_A$) and with interactive SST ($\Delta_O$). Differences not significant at the 0.05 level are hatched gray.

tent with previous climate model simulations showing the influence of land surface forcing on tropical circulation. Including vegetation on the land surface strengthens ITCZ, monsoon, and Hadley cell dynamics, as well as intensifying and maintaining the global water cycle, compared to a desert planet (Svirezhev and von Bloh, 1998; Fraedrich et al., 1999; Cresto Aleina et al., 2013; Rombouts and Ghil, 2015). Further, in a previous version of the GISS GCM, implement-

ing an improved representation of vegetation stomatal conductance and photosynthesis dependence on atmospheric humidity and $CO_2$ concentration decreased biases in precipitation over the oceanic ITCZs and tropical South America (Friend and Kiang, 2005). More specifically, afforestation in the northern midlatitudes shifts the ITCZ northward (Swann et al., 2012), while deforestation in northern middle and high latitudes shifts the ITCZ south (Devaraju et al., 2015). This

# Sea Surface Temperature (K), $\Delta_O$

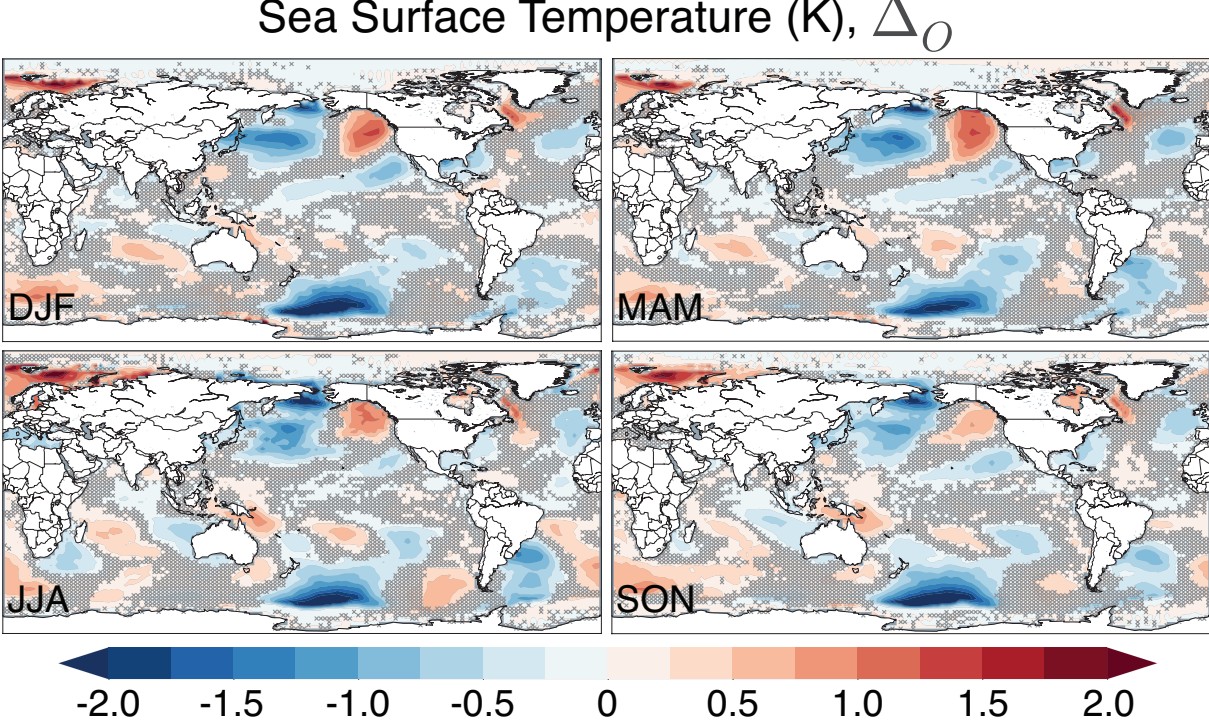

**Figure 4.** Difference in SST (K) due to irrigation with interactive SST ($\Delta_O$). Differences not significant at the 0.05 level are hatched gray.

non-local climate impact of land cover implies that expanded forest cover in Eurasia might explain the wetter conditions in northern Africa inferred for the mid-Holocene (Swann et al., 2014). In our experiments, irrigation under interactive SST results in tropical precipitation decreasing in the Northern Hemisphere and increasing in the Southern Hemisphere (Figure 7), consistent with irrigation (like boreal deforestation in Devaraju et al. (2015)) exerting its main cooling effect on the Northern Hemisphere and thus increasing northward heat transport and shifting the Northern Hemisphere Hadley cell southward. Our modeling results confirm those of Swann et al. (2012), who note that "Interaction between sea surface temperatures and the atmosphere is necessary for allowing shifts in large-scale circulation and precipitation."

The work reported here has several important limitations. Our q-flux simulations gave an equilibrium impact of the irrigation forcing on climate. However, in reality, ocean circulation and mixing delay equilibrium with forcings such as irrigation. Since irrigation has only been practiced globally at its current magnitude for the past few decades, it is expected that transient changes in SST due to irrigation for the current climate system would be smaller than the equilibrium changes simulated here. On the other hand, allowing changes in ocean currents and heat transport could possibly also enhance climate impacts compared to our q-flux configuration (which had effectively constant ocean heat transport). Preliminary comparison of SSTs in irrigation and no-

irrigation runs of GISS ModelE2 with time-varying forcings and a three-dimensional dynamic ocean model (Cook et al., 2015) suggests that around the year 2000, the amplitude of non-local SST changes due to irrigation might have been ∼0.1-0.2 K, instead of the ∼0.5-1 K seen here with a q-flux model run to equilibrium. These differences between transient and equilibrium responses to LCLUC in the coupled atmosphere-ocean system should be explored in more detail.

Future changes in irrigation are highly uncertain (Wada et al., 2013; Elliott et al., 2014), particularly given the depletion of groundwater sources of irrigation water in many major agricultural areas (Wada et al., 2012; Gleeson et al., 2012; Krakauer et al., 2013; Leng et al., 2014). As well, water diversion for irrigation impacts riverine freshwater fluxes and sea level (Chao et al., 2008; Wisser et al., 2010), which may in turn affect climate in ways not represented in our simulations.

The present work only shows that interactive SST alters the climate effects of irrigation in GCM simulations, by generating SST anomalies that extend irrigation's impact spatially and temporally. Simulated changes with interactive SST are, in principle, more physically consistent than those simulated under fixed SST in that energy is being conserved, though the constraints of the q-flux surface ocean can also introduce biases. We have not conducted comparisons with observations to directly verify that simulations with interactive SST actually represent irrigation effects on climate

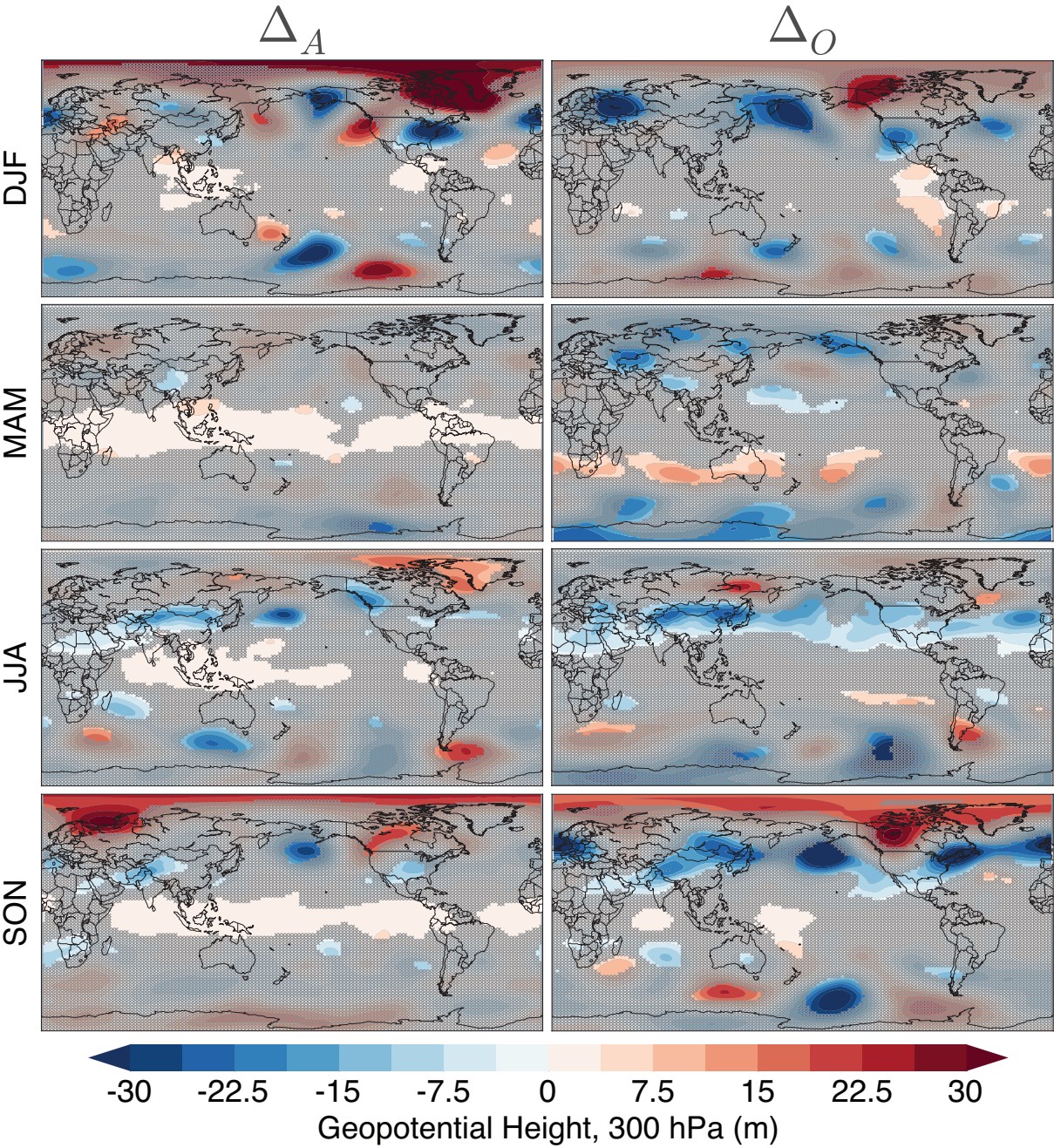

**Figure 5.** Same as Figure 3, but for 300-mb height differences (m).

better than simulations with fixed SST. Showing that the responses seen with interactive SST are consistent across different GCMs could increase confidence that the results reported here are physically meaningful and not an artifact of particular model configurations. If this turns out to be the case, the extensive non-local changes in patterns of SST and other climate variables seen in our simulations suggest that studies of irrigation climate impacts that use either global models with fixed SST or regional models with fixed boundary conditions (Im et al., 2014; Alter et al., 2015) may miss some of the impact of irrigation on non-local climate.

## 5   Conclusions

We compared simulations of the equilibrium effect of contemporary irrigation extent on climate with and without interactive sea surface temperatures to show that, in these

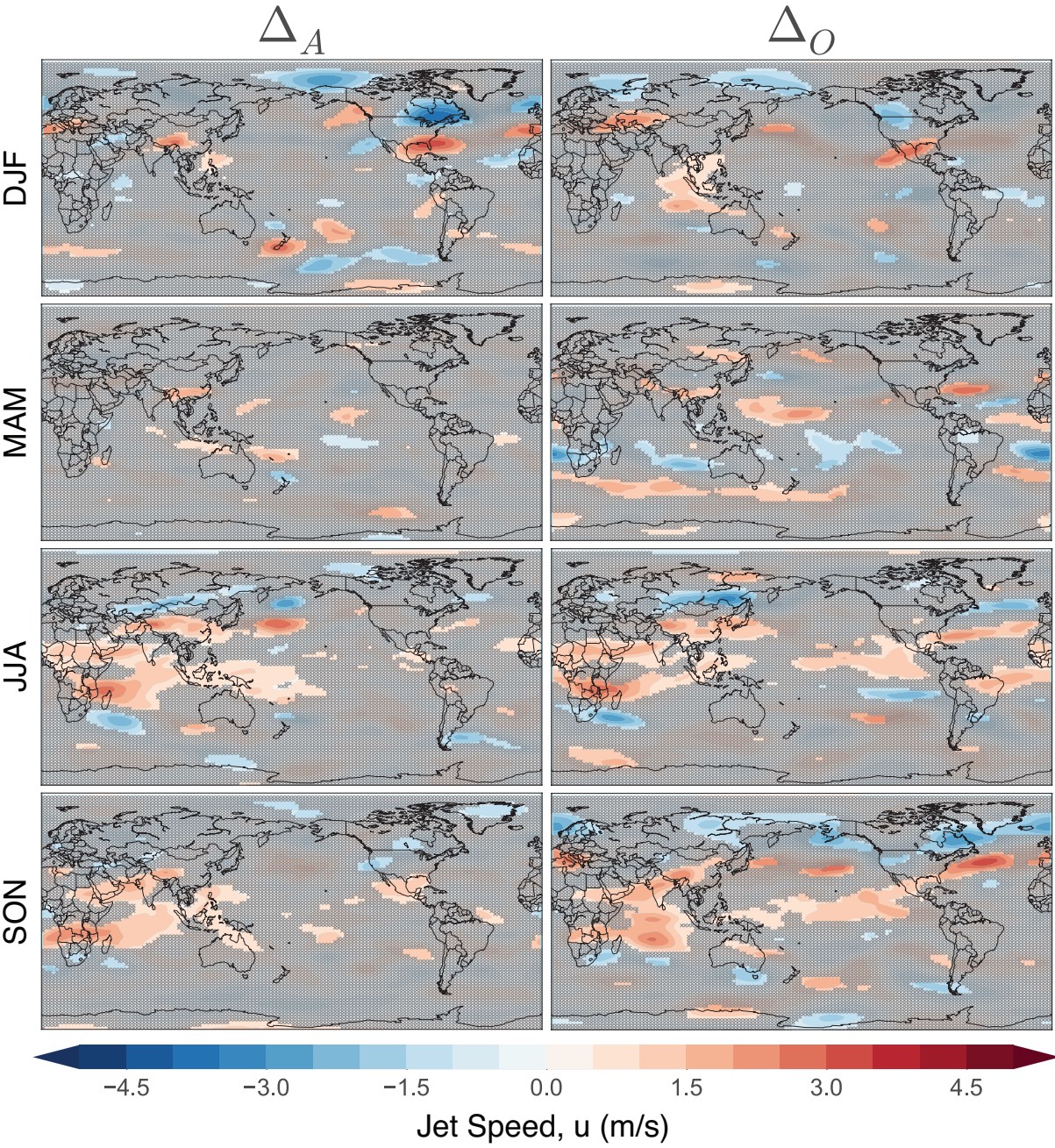

**Figure 6.** Same as Figure 3, but for jet stream meridional velocity differences (m s$^{-1}$).

simulations, air-sea interaction does impact the magnitude of global-mean and spatially-varying climate impacts and greatly increase their global reach. Air-sea interaction amplified irrigation-driven standing wave patterns in the tropics and midlatitudes, approximately doubling the global mean amplitude of surface air temperature changes due to irrigation. Subject to confirmation with other models and consideration of irrigation's time evolution, these findings imply that LCLUC may be an important contributor to climate change even in remote areas such as the Southern Ocean, and that attribution studies need to consider LCLUC such as irrigation as truly global forcings that affect climate and the water cycle in ocean as well as land areas.

### Code and data availability

The GISS GCM source code can be accessed from http://www.giss.nasa.gov/tools/modelE/ for free download and

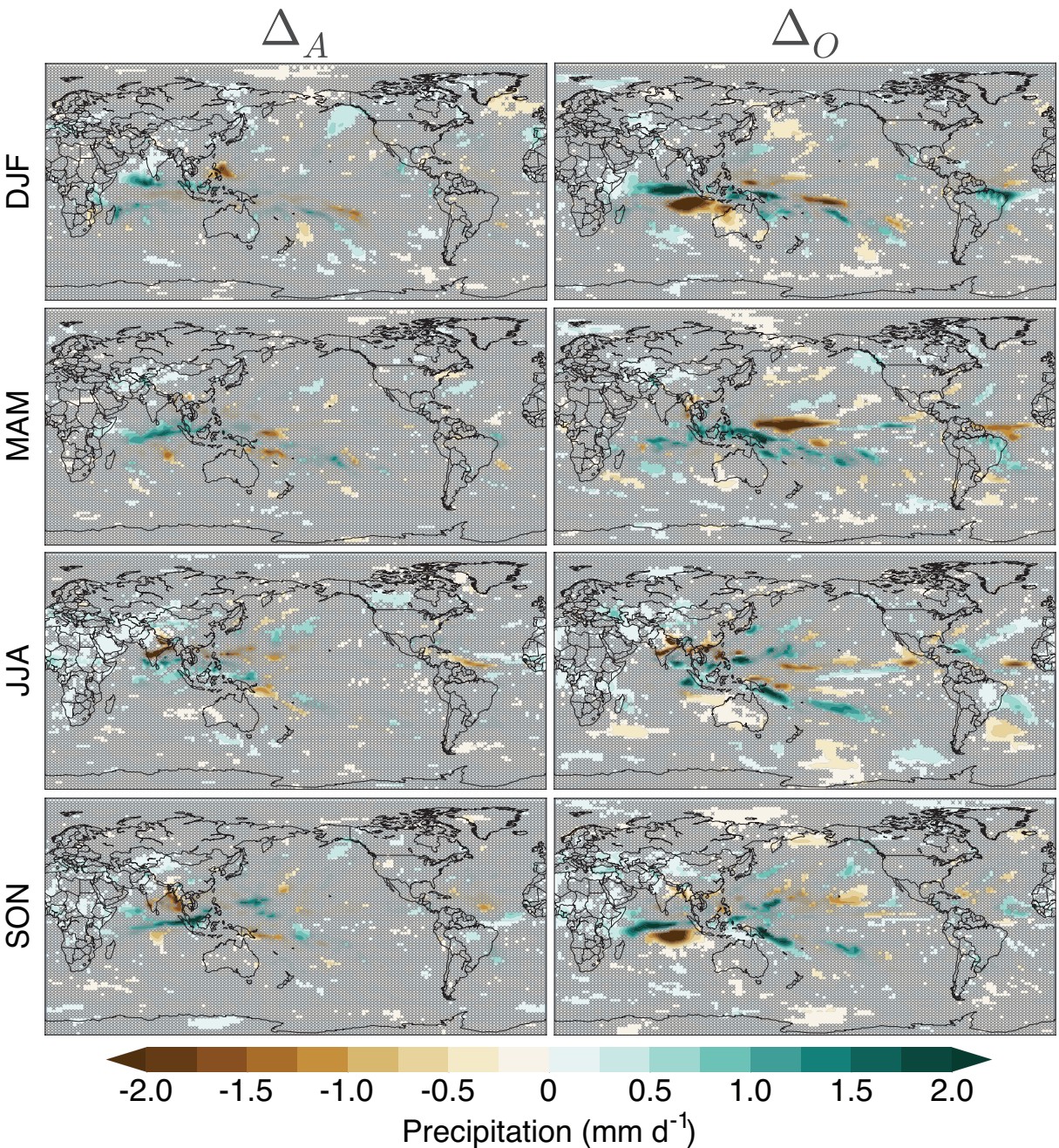

**Figure 7.** Same as Figure 3, but for precipitation differences (mm day$^{-1}$).

use. Documentation of model configurations and further references are also available there.

**Acknowledgements.** Climate modeling at GISS is supported by the NASA Modeling, Analysis, and Prediction program. Resources supporting this work were provided by the NASA High-End Computing (HEC) Program through the NASA Center for Climate Simulations (NCCS) at Goddard Space Flight Center. The authors specifically thank Maxwell Kelly for assistance with the model irrigation module and output diagnostics. M.J. Puma gratefully acknowledges support from the Interdisciplinary Global Change Research under NASA cooperative agreement NNX14AB99A supported by the NASA Climate and Earth Observing Program and from the Columbia University Center for Climate and Life, where he is a Climate and Life Fellow. NYK gratefully acknowledges support from NOAA under grants NA11SEC4810004, NA12OAR4310084, and NA15OAR4310080; from CUNY through PSC-CUNY Award 68346-00 46 and CUNY CIRG Award 2207; and from USAID

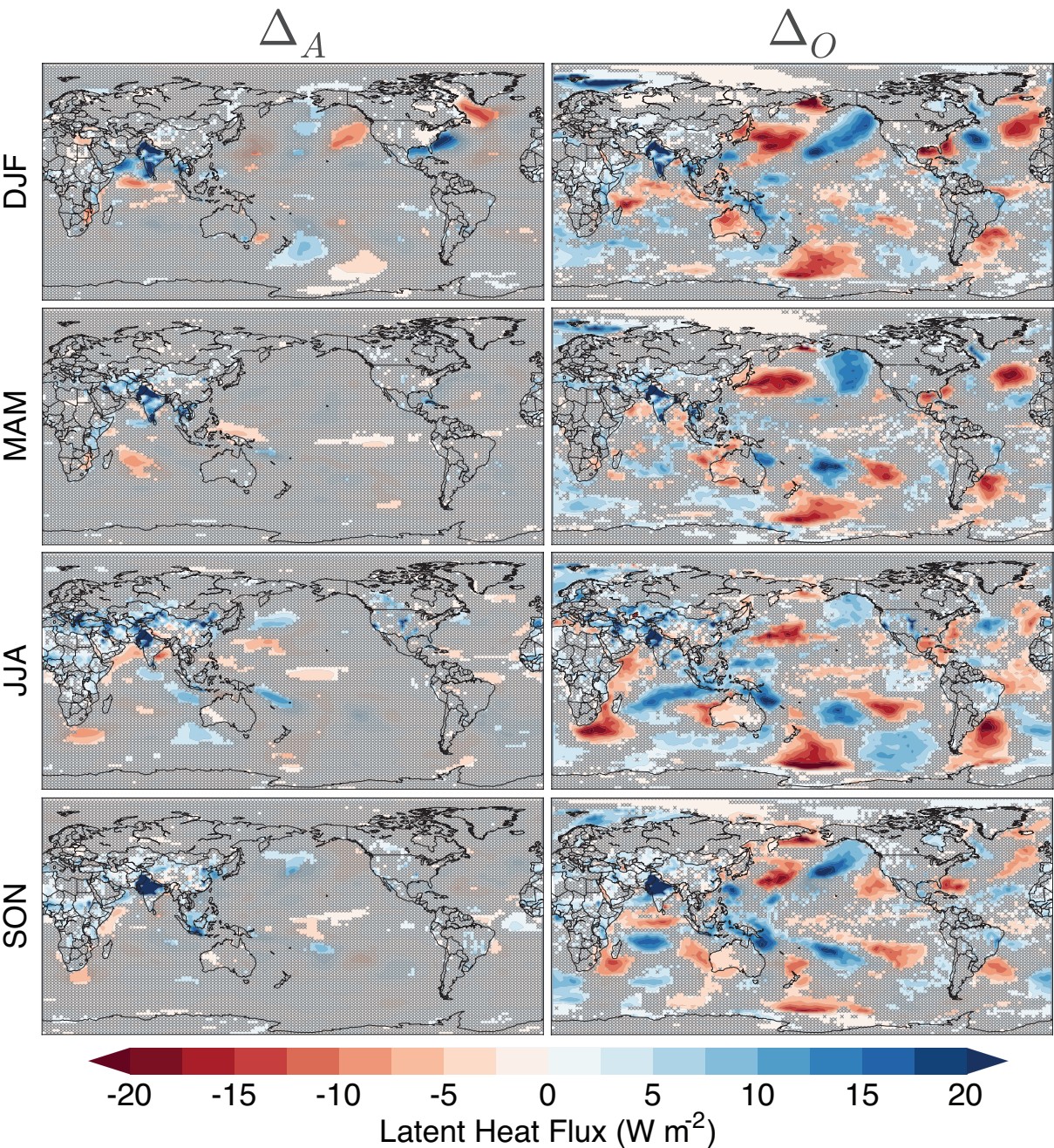

**Figure 8.** Same as Figure 3, but for surface latent heat flux differences (W m$^{-2}$).

IPM Innovation Lab award "Participatory Biodiversity and Climate Change Assessment for Integrated Pest Management in the Annapurna-Chitwan Landscape, Nepal". All statements made are the views of the authors and not the opinions of the funding agency or the U.S. government.

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
