# Peer review of "Ocean-atmosphere interactions modulate irrigation's climate impacts"

_Earth System Dynamics, 2016_

## Referee Comment (RC1) · Anonymous Referee #1 · 13 Jun 2016

Anonymous reviewer recommendation – "Ocean-atmosphere interactions modulate irrigation's climate impacts"

This study uses ModelE2 to explore the irrigation's impacts with and without considering ocean- atmosphere interactions via using a slab ocean model. They found that the ocean-atmosphere interaction amplifies irrigation-driven climate responses in the tropics and midlatitudes, approximately doubling the global mean amplitude of surface temperature changes from irrigation. In general, irrigation's impacts have been gained lots of attentions recently by using different numerical models both at regional and global scale. However, most of the previous studies used prescribed SST as the boundary forcings. This study pioneers using the slab ocean to explore how the ocean-atmosphere interaction can modulate the irrigation's impacts. While this study points interesting irrigation's remote effects, the analysis in this study could be improved in or-

der to support what the authors want to convey. Overall, the findings presented in this paper may be of interest to the community. There are several aspects that the authors may want to consider in the revised version. Please see below for some comments.

1. What is the overall performance of GISS-ModelE2? The global pattern of precipitation/circulation compared to reanalysis data may be useful.

2. The authors have shown the spatial patterns for surface temperature, precipitation, and pressure from the effects of irrigation. How the circulation changes when applying the irrigation?

3. The authors argue that the patterns induced by irrigation forcing are less widespread when SST is fixed, implying that ocean-atmosphere interaction is a key to their propagation and persistence across seasons. However, how do we know the persistence can across the seasons?

4. While the authors find an interesting finding of enhancing the wave patterns from the irrigation forcing when considering the ocean-atmosphere interactions, how about the time series analysis? How does LCLUC affect the interannaul variations? Also, will including the ocean-atmosphere interaction affect such variations?

5. The authors use a slab-ocean (without the ocean dynamics) to show the remote effects when considering the ocean- atmosphere interactions. Can the authors comment on what the remote impacts might change if using a fully coupled ocean dynamic model?

6. Why is there some remote impacts over the southern oceans when considering the ocean-air interactions? Wave patterns propagations?

7. Do the simulations reach the equilibrium yet? A plot for the energy balance in TOA might be worth.

8. There is some northward shift of the ITCZ during the winter. Is there any particular reason for this?

---

## Referee Comment (RC2) · Anonymous Referee #2 · 13 Jun 2016

**1    General comments**

The present article addresses a research question that, as of late, is being investigated by an increasing number of studies, i.e. irrigation as a man made land cover change that substantially alters the (global) climate. Many of these studies rely on a range of climate models, and the present study presents a valuable contribution by demonstrating how the setup of these models (treatment of the ocean) can affect the simulated impact of irrigation. As the authors, state there has been a number of studies, both with and without interactive SSTs, but to the best of my knowledge, no study investigates the respective effects in detail. Here, the authors do a very good job presenting their motivation and placing their study within the existing body of literature, giving a concise overview of recent studies.

The authors address the issue of interactive vs. prescribed SSTs by running the GISS model in a setup with a slab-ocean and compare the simulated irrigation induced impacts to simulations in which SSTs were prescribed. This approach is very intuitive and, in my opinion, the authors choose a good approach, by keeping the introduction short, but providing sufficient information about the model and the setup.

The result section of the article is well structured and written using appropriate language. There are only few sentence that required a second reading and these could be fixed with a minimum effort (see point by point). Additionally, the figures and tables are well compiled, making it very easy to follow the authors.

However, I am concerned with one of the conclusions the authors appear to draw from their results, namely, that simulations, with an interactive ocean model are better suited to represent ocean-atmosphere interactions and are therefore superior to those with prescribed SSTs. This conclusion has substantial implications for anyone who will investigate irrigation related impacts using a climate model, as it would basically disqualify studies using prescribed SSTs. I am not convinced that the methods/ results suffice to make such a substantial claim. Nonetheless, despite this disagreement on the conclusion, my opinion is that the authors present a well constructed study that provides instrumental insights into an important topic.

**2   Specific comments**

With respect to the method section of the article, I thought that it was appropriate in length and style and I generally liked that it was very concise. However, I think it would be beneficial if the authors could provide some form of measure or graph,

demonstrating to the reader that the model is in fact in equilibrium after 10 years of spin-up time. The authors state that, if possible, irrigation requirements are satisfied from the river discharge. This should have a substantial impact on the land-surface hydrology and by that possibly on the ocean as well. This is especially relevant because in the discussion section the authors give an interesting comparison to another GISS-based simulation in which the system was not in equilibrium. Here, they speculate that the differences in the ocean's response to the irrigation forcing could be related to the fact that in the present study an equilibrium response was investigated whereas the other study looked at a transient response (and not to the fact that one study used a slab-ocean and the other a fully dynamic 3-d model).

Furthermore, maybe the authors could give a brief reasoning for why they used the slab-ocean instead of a fully dynamic 3-d model (as I understand, some ocean modellers would claim that the 3-d model is a better representation of reality)?

As stated above, my only real concern is that, from the results, I do not arrive at the same conclusions as the authors (that simulations, with an interactive ocean model do a better job at capturing ocean-atmosphere interactions and are therefore superior to those with prescribed SSTs). An example for this conclusion is the claim at the end of the abstract, i.e. that (simulation-based) attribution studies should include interactive oceans. This I find questionable for the following reasons:

It can be argued that an interactive ocean component introduces additional variability and uncertainty into the model. Thus, by prescribing SSTs (if these can be considered to be reliable) the model is actually constrained to a more realistic representation of present day climate than a model with an interactive ocean.

But more importantly, the authors compare individual simulations, not ensembles, and it is debatable whether a ttest is a sufficient tool to evaluate if differences are related to changes in the model physics. Often even slight changes (not even in the physics), e.g.

in the initial conditions, are sufficient to cause the distributions in two simulations to be statistically significantly different. Taking this into account, the differences between $Delta - A$ and $Delta - O$ on the land surface appear to be quite small, in figures 2,5 and 6. This is also indicated by the mean differences $Delta - Delta$ for most variables not being statistically significant.

Thus, another possible way to interpret the study's results is that, with respect to the land surface, simulations with a model setup with a slab-ocean are quite comparable to those forced with prescribed SSTs. This shows that the irrigation induced impacts are persistent and so strong that they are not concealed by the (ocean) model's internal variability. Additionally it indicates that a model configuration with an interactive (slab-) ocean is a very suitable tool to investigate irrigation related climate impacts in the future when reliable SST-data may not be obtainable.

If the authors maintain the claim that the statistically significant differences in the simulated irrigation impact are due to atmosphere ocean feedbacks (that can only be accounted for with an interactive ocean model), I would urge them to demonstrate this using ensemble simulations. If it is not feasible to conduct ensemble simulations, it would greatly help if the authors could at least perform one additional simulation with an interactive ocean and slightly altered initial conditions. With this simulation they could show that there are no statistically significant differences between two simulations that are based on the same model setup but using slightly different initial conditions. This would give some confidence that the significant differences they show are related to the ocean atmosphere feedbacks, even though I think ensemble simulations would be the preferable approach.

Furthermore, for the claim that simulations (for the present day) using prescribed SSTs may miss important effects, it would be very helpful if they could give some indication for these effects also existing in the real world, e.g via a comparison to observations of

e.g. precipitation, surface temperatures or the wind field.

**3  Point by point**

- P. 1, l. 3: In the title it is "irrigation's climate impacts"?

- P. 1, l. 3: Maybe better "... contemporary irrigation (the geographic extent and irrigation intensity correspond to those of the year 2000) ..."

- P. 1, l. 9,10: as stated above, from the results presented, I do not arrive at this conclusion, (that attribution studies should include an interactive ocean).

- P. 2, l. 5: Maybe better "... to persist and to be transferred between ...".

- P. 2, l. 5: Maybe better "... the irrigation related climate forcing ...".

- P. 2, l. 22 - 24: Would it be possible to convert these values to km$^3$/a as this would make it easier to compare them with other studies ?

- P. 3, l. 1 - 2: Would it be possible to give more information on how these 10 years were determined?

- P. 3, l. 3: Maybe better "For the climate variables of interest ...".

- P. 3, l. 5: Maybe better " ... interactive SSTs ...".

- P. 3, l. 7: Maybe better "... using a Student's ...".

- P. 4, l. 15: Maybe better "... that directly quantify the conditions and the moisture status at earth's surface ...".

- P. 4, l. 7: Maybe better " The irrigation-induced ... over irrigated areas, spreads ...".

- P. 4, l. 12: Maybe better ". Irrigation-induced changes in the surface latent and sensible heat fluxes ...".

- P. 4, l. 18: I think the information in the brackets is not required as the terms SST and soil moisture already imply the geographic location.

- P. 4, l. 18: Maybe better ". Over land, the cooling ...".

- P. 5, l. 19: Maybe the sentence could be split up. At the moment it reads as if the mean amount would refer to the cooling.

- P. 4, l. 20: Maybe better ". Over the ocean, the cooling ...".

- P. 4, l. 23: Reading the sentence I was wondering whether I had overlooked the zonal means. As they are not shown maybe its better to just refer to the global mean.

- P. 6, l. 2: I find this difficult to see in the figure. To me it appears that over land areas the patterns of pronounced impacts especially in Southern Asia are actually quite comparable. Maybe an irregular spaced colorbar could be helpful to see differences between 0.4 and 0.8 K.

- P. 6, l. 12: The wave patterns are not exclusiv for the q-flux simulations, but there is also a wave pattern present for fixed SSTs in the Southern Hemisphere in JJA.

- P. 7, l. 15 - 16: Maybe better "... air-sea interactions ... the divergence in the irrigation responses (surface air temperature and geopotential height) between ...".

- P. 7, l. 17 - 18: Maybe better "... with the phases shifted between the interactive SST and fixed SST simulations ...".

- P. 7, l. 6 - 7: Maybe better "... study using a different atmosphere and land surface model and found that ...".

- P. 7, l. 10 - 15 : This is possibly true, but just as likely the differences are not related to the model physics. This is very hard to tell from comparing individual simulations.

- P. 7, l. 27: Maybe better "... patterns are less pronounced ...".

- P. 9, l. 33: Maybe better "... to illuminate the ... an to identify ...".

- P. 9, l. 2: Maybe better "... the irrigation forcing ...".

- P. 9, l. 2 - 4 : Here, it is true that the simulations with slab-ocean are energy conserving and thus more physics-based, but at the same time there is additional uncertainty that could lead to simulations with a slab-ocean to be further from reality than those with fixed SSTs. As in the following the authors discuss how the simulations may compare to the real world I think this could also be mentioned at this point.

- P. 13, last sentence section 4.: Again, this is possibly true, but just as likely the differences found in this study are not related to the model physics. This is very hard to tell from comparing individual simulations.

- P. 13, l. 3 - 4 : Is this the surface air temperature? Does this mean include the ocean?

- With respect to figure 2, I just had slight difficulties to clearly see the differences between 0.4 and 0.8 Kelvin that the authors discuss on page 6 line 30 ff. Maybe a

slight alteration of the colorbar (maybe irregular intervals ?) could make it easier to identify these differences.

- With respect to the tables, would it be possible to also include the value of $Delta - Delta$ ? Maybe the authors could also give an indication of significance for $Delta - A$ and $Delta - O$? I think this would make it even easier for the reader to get a feeling of the importance of $Delta - Delta$ relative to $Delta - A$ and $Delta - O$.

---

## Referee Comment (RC3) · Anonymous Referee #3 · 17 Jun 2016

1) The question that motivates the study is very interesting, but I think that the authors overstate their case too much to be convincing in their conclusions. My overall opinion is that this paper, although dealing with a novel and interesting question, is too modest in its present state. The simplicity of the numerical design and the performed analyses suggest the authors overlook the complexity arising from coupling land, atmosphere and ocean in climate models, which is problematic for publishing in "Earth System Dynamics".

2) My main concern is that the reported differences between the interactive and fixed SST runs are weak and moderately significant, both at the large scale (in term of p-value in Table 1), and over the maps, in which the areas with insignificant changes are much larger than the ones with a significant change. The main exception is the SSTs themselves, but this is not very informative given that their variability is very different

by construction in the two kinds of experiments (see also my comments a-b below).

Most of the recent papers that deal with tiny changes against the internal variability of the climate system use an ensemble approach to be more convincing from a statistical point of view, and I would like the authors explaining why they did not do the same.

Even if we accept that the comparison of single members for each experiment is justified, information is missing in the paper regarding the experiment design and the subsequent statistical analysis:

a) I understood that the fixed SST simulations were analyzed over 50 years, while the SST forcing is available over 9 years only (1996-2004): how is it done, by cycling the 9-yr forcing over the 50 years? If so, it implies a very different variability to the one of the interactive SST simulations, for which we also need to know if you impose or not an increasing amount of GHG and aerosols, as this may induce a trend in addition to the seasonal to inter-annual variability (I'm not a specialist of slab oceans, so I need that kind of information to make sense of the results). I also wonder why the fixed SST runs do not rely more strongly on the AMIP protocol, which comes with a much longer SST forcing data set, starting in 1979.

b) The above information is important since the significance of the analyzed differences is tested based on Student's t-test, which basically compares the mean difference to the variability (standard deviation) of the two compared samples. Regarding the test, I did not understand what was behind the following mention (p3, L8) "with the degrees of freedom adjusted based on the lag-1 autocorrelation of the time series". Since the significance depends on the degrees of freedom, I recommend clarifying this point in the paper.

c) The paper relies on comparing the effect of irrigation in fixed and interactive SST experiments. These effects are respectively called $\Delta_A$ and $\Delta_O$, and calculated as the difference between an irrigated and control experiment, in each of the confirmations. The rationale is that if there is a significant difference between $\Delta_A$ and $\Delta_O$, then it means that the interactive SST influences the response of the climate to irrigation. But

we may imagine another explanation to a significant difference between $\Delta_A$ and $\Delta_O$, because the two control simulations must be different (AMIP and CMIP simulations are different), and may drive the differences between $\Delta_A$ and $\Delta_O$. Thus, I think the authors need to compare the differences between the two control experiments and the ones between $\Delta_A$ and $\Delta_O$ before concluding anything.

3) I also regret that the analysis of the changes and the attempt to give them a physical explanation is rather superficial. The studied changes are induced by enhanced moisture input to the atmosphere over irrigated land, and the atmospheric humidity is not analyzed. The only circulation variable is the 300-mb height, and no mention is made to moisture convergence and convection for the atmospheric compartment, nor to monsoons and surface ocean currents. Yet, if there is an influence of the interactive ocean on the response of the climate to irrigation, it should imply that irrigation changes the ocean's behavior between the two interactive SST runs.

4) An illustrative example of the overstatement and lack of physical insight that can be found throughout the paper is the analysis of the precipitation changes over eastern Africa. We are asked to compare the $\Delta_A$ and $\Delta_O$ in precipitation in MAM over eastern Africa in Fig. 5, but there is almost nothing! I'm not even sure there would be something discernable with a magnifying glass. This extremely weak change receives the longest explanation of the entire paper, with a 10-line paragraph, but it ends with a rather weak and speculative conclusion (p7, L13-15): "Thus, ocean-atmosphere interactions may importantly affect the magnitude and location of non-local irrigation impacts on climate, such as those potentially implicated in precipitation trends in eastern Africa."

5) Minor comments:
p2, L22: summed should probably be replaced by averaged
p3, L10 and p4, L24: do you analyze the rms or the standard deviation? The latter seems more informative, as it excludes the effect of differing means.
P4, L14: when analyzing large scale means (land vs ocean), it's abusive to write that "interactive SST spreads the cooling": you need maps to draw this conclusion.
Caption of Table 2 should mention that the values correspond to the northern hemisphere only.

Caption of Table 3 should mention that the studied variability (by means of rms) is spatial (and not temporal, which could be worth an analysis too).

P7, 1st L15-16: please clarify "supporting the role of air-sea interaction in driving the divergence in surface air temperature and geopotential height irrigation responses between the fixed-SST and interactive-SST simulations."

P7, L17-18: please clarify what is really meant by "the same stationary wave pattern [...] is found [...], with shifted phase"

P7, L5 (according to the numbering in the pdf): please clarify "The role of interactive SST in non-local irrigation climate forcing"

---

## Referee Comment (RC4) · Anonymous Referee #4 · 3 Jul 2016

General comments:

The manuscript presents a study on the role of ocean-atmospheric interactions in modulating the impacts of irrigation on large scale climate.

By linking the atmosphere with the ocean and the land, the paper is within the scope of ESD. However, the manuscript would benefit if the authors would treat the interactions between the modelled components from a more physical perspective and thoroughly discuss the processes and complexities behind the numerical experiments that lead to the differences presented.

I think that with a little expansion of the scope/details of the study the authors could produce a much more informative and well-rounded paper compared to the current, rather short, 'letter like' manuscript. Overall, the methods are described very briefly. I

suggest the authors to further elaborate some of the details in the main text instead of just referring to previous studies (e.g. for the irrigation scheme and the model runs (see also specific comments below)).

Authors should investigate/comment of the effects on the oceans of taking all the water from the rivers. I.e. less water reaching the ocean and the effect on ocean temperatures.

The analysis is performed with a focus on spatial impacts, but I think the study would benefit if also some more attention would be given to the temporal effects. I.e. do the impacts change between the different years that were simulated and if so what are the main reasons?

My final concern is that only one model with each of the configurations is being run and analysed, but little consideration is given to the uncertainties associated with the different initial conditions and model configurations. The authors should at least test/discuss if the effects observed from different initial states are larger than the effects one obtains from the presence/absence of irrigation and/or interactive SST.

Based on the comments above and below, I suggest reconsidering the manuscript for publication after major revisions

Specific comments:

P1L19: please elaborate on the factors that prevent the deduction of remote impacts based on observations.

P2L7: Please specify the degree of amplification, i.e. was the amplification significant?

P2 1st Paragraph: Were any major reservoirs considered before water was taken from the groundwater resources? Were the volumes of water and groundwater abstractions of the same magnitude as actual measured irrigation? In many countries where irrigation was applied, rice is one of the major crops grown. Have the effects of the cropping methods (e.g. rice paddies result in standing water and therefore into additional standing water and evaporation) been considered in the model? Please comment/elaborate on these points in the manuscript.

P3L1: the simulations were run for 60 years, however the data available for the SST and the seas ice were only computed for 1996-2004. Does this short time span cover the natural variability of these variables? Please elaborate.

P3 Fig1: I suggest using a different colour scale (not starting with white), as it is difficult to distinguish if for an area no/ little irrigation has been applied. Additionally, please elaborate why certain areas have high amounts of irrigation applied, although it is the season of high precipitation (e.g. India during the Monsoon).

P3L8: Please elaborate why a lag-1 adjustment was performed. What is the physical meaning behind this? Does the system not exhibit any long term memory?

P3L12-15: Why only northern Hemisphere? I would think that the other 8% of irrigation also merits consideration.

P7L5-10: I would suggest moving this to the discussion section.
* * *

---

## Author Comment (AC1) · 5 Aug 2016

We thank the reviewers for their thoughtful reading, and have prepared a revised and improved version of our paper. Our main changes include adding more details on the simulation configurations and on the statistical analysis, adding a figure to show the model spin-up, adding more significance tests to the tables, showing more clearly the circulation changes seen, and removing a paragraph on relatively small precipitation changes seen in Africa.
Our responses to each point raised are as follows:

**1 Review 1:**

*1. What is the overall performance of GISS-ModelE2? The global pattern of precipitation/circulation compared to reanalysis data may be useful.*

The cited reference [1] evaluates the ability of GISS-ModelE2 to simulate observed climate fields. We now added another reference [2] to a comprehensive study of the ability of GISS-ModelE2 to simulate trends in precipitation, circulation, temperature, and sea ice, compared to observations.

*2. The authors have shown the spatial patterns for surface temperature, precipitation, and pressure from the effects of irrigation. How the circulation changes when applying the irrigation?*

We now added maps of change in the jet stream speed (as another indicator of atmospheric circulation) along with description of these results.

*3. The authors argue that the patterns induced by irrigation forcing are less widespread when SST is fixed, implying that ocean-atmosphere interaction is a key to their propagation and persistence across seasons. However, how do we know the persistence can across the seasons?*

We now removed this phrase.

*4. While the authors find an interesting finding of enhancing the wave patterns from the irrigation forcing when considering the ocean-atmosphere interactions, how about the time series analysis? How does LCLUC affect the interannual variations? Also, will including the ocean-atmosphere interaction affect such variations?*

Effects of soil moisture boundary condition on time variation and autocorrelation of climate variables have been explored in previous work with ModelE [3]. In the current paper, we focus primarily on annual and seasonal mean climate differences due to irrigation under fixed SST versus slab ocean.

*5. The authors use a slab-ocean (without the ocean dynamics) to show*

*the remote effects when considering the ocean-atmosphere interactions. Can the authors comment on what the remote impacts might change if using a fully coupled ocean dynamic model?*

Yes, this is mentioned in the Discussion, with reference to [4]. For the present study, we conducted experiments of steady-state irrigation impacts using either fixed SST or a slab ocean because that configuration reaches an equilibrium state within a few simulation years, whereas a fully coupled ocean dynamic model would take thousands of model years to reach equilibrium under a particular pattern of irrigation forcing.

*6. Why is there some remote impacts over the southern oceans when considering the ocean-air interactions? Wave patterns propagations?*

Yes, we now mention this in the Discussion.

*7. Do the simulations reach the equilibrium yet? A plot for the energy balance in TOA might be worth.*

We now added a figure showing the time evolution of the TOA energy balance and surface temperature in the 4 runs to show the approach to equilibrium.

*8. There is some northward shift of the ITCZ during the winter. Is there any particular reason for this?*

We in fact find changes in precipitation over all seasons throughout much of the deep tropics. Exploring the robustness of these results and the generating mechanisms could be a worthwhile target for follow-up work.

**2   Review 2:**

a) *P. 1, l. 3: In the title it is "irrigation's climate impacts"?*

Yes.

b) *P. 1, l. 3: Maybe better ". . . contemporary irrigation (the geographic extent and irrigation intensity correspond to those of the year 2000) . . . "*

Modified.

c) *P. 1, l. 9,10: as stated above, from the results presented, I do not arrive at this conclusion, (that attribution studies should include an interactive ocean).*

We reworded the conclusion to be more cautious.

d) *P. 2, l. 5: Maybe better ". . . to persist and to be transferred between . . . ".*

*P. 2, l. 5: Maybe better "...the irrigation related climate forcing ...".*

Modified.

e) *P. 2, l. 22 - 24: Would it be possible to convert these values to $km^3/a$ as this would make it easier to compare them with other studies ?*

Yes, this is now added.

f) *P. 3, l. 1 - 2: Would it be possible to give more information on how these 10 years were determined?*

We now clarify that we use the average of those years as a basis for simulating 'contemporary' (around year 2000) climate.

g-l) *P. 3, l. 3: Maybe better "For the climate variables of interest ...".*

*P. 3, l. 5: Maybe better "...interactive SSTs ...".*

*P. 3, l. 7: Maybe better "...using a Student's ...".*

*P. 4, l. 15: Maybe better "...that directly quantify the conditions and the moisture status at earth's surface ...".*

*P. 4, l. 7: Maybe better " The irrigation-induced ...over irrigated areas, spreads ...".*

*P. 4, l. 12: Maybe better ". Irrigation-induced changes in the surface latent and sensible heat fluxes ...".*

We checked these and modified the phrasing as appropriate.

m) *P. 4, l. 18: I think the information in the brackets is not required as the terms SST and soil moisture already imply the geographic location.*

Admittedly not required, but we believe it is worth keeping as confirmation to the reader.

n) *P. 4, l. 18: Maybe better ". Over land, the cooling ...".*

Changed.

o) *P. 5, l. 19: Maybe the sentence could be split up. At the moment it reads as if the mean amount would refer to the cooling.*

Yes, that is the intention.

p) *P. 4, l. 20: Maybe better ". Over the ocean, the cooling ...".*

Changed.

q) *P. 4, l. 23: Reading the sentence I was wondering whether I had overlooked the zonal means. As they are not shown maybe its better to just refer to the global mean.*

Changed to "global or Northern Hemisphere"

r) *P. 6, l. 2: I find this difficult to see in the figure. To me it appears that over land areas the patterns of pronounced impacts especially in Southern Asia are actually quite comparable. Maybe an irregular spaced colorbar could be helpful to see differences between 0.4 and 0.8 K.*

We modified the color scheme to show this difference more clearly. The typical numerical values of changes are also shown in Table 3.

s) *P. 6, l. 12: The wave patterns are not exclusive for the q-flux simulations, but there is also a wave pattern present for fixed SSTs in the Southern Hemisphere in JJA.*

We now clarified that the wave pattern is much more pronounced for the q-flux simulations, though it is present to some extent with fixed SST also.

t) *P. 7, l. 15 - 16: Maybe better ". . . air-sea interactions . . . the divergence in the irrigation responses (surface air temperature and geopotential height) between . . . ".*

Changed.

u) *P. 7, l. 17 - 18: Maybe better ". . . with the phases shifted between the interactive SST and fixed SST simulations . . . ".*

Changed.

v) *P. 7, l. 6 - 7: Maybe better ". . . study using a different atmosphere and land surface model and found that . . . ".*

We removed this paragraph.

w) *P. 7, l. 10 - 15 : This is possibly true, but just as likely the differences are not related to the model physics. This is very hard to tell from comparing individual simulations.*

We removed this paragraph.

x) *P. 7, l. 27: Maybe better ". . . patterns are less pronounced . . . ".*

Changed.

y) *P. 9, l. 33: Maybe better ". . . to illuminate the . . . an to identify . . . ".*

Clarified.

z) *P. 9, l. 2: Maybe better "... the irrigation forcing ...".*

Changed.

aa) *P. 9, l. 2 - 4 : Here, it is true that the simulations with slab-ocean are energy conserving and thus more physics-based, but at the same time there is additional uncertainty that could lead to simulations with a slab-ocean to be further from reality than those with fixed SSTs. As in the following the authors discuss how the simulations may compare to the real world I think this could also be mentioned at this point.*

Agreed, we modified our phrasing to better reflect this.

ab) *P. 13, last sentence section 4.: Again, this is possibly true, but just as likely the differences found in this study are not related to the model physics. This is very hard to tell from comparing individual simulations.*

True. We rephrased this sentence to be more specific.

ac) *P. 13, l. 3 - 4 : Is this the surface air temperature? Does this mean include the ocean?*

Yes, we clarify this now.

ad) *With respect to figure 2, I just had slight difficulties to clearly see the differences between 0.4 and 0.8 Kelvin that the authors discuss on page 6 line 30 ff. Maybe a slight alteration of the colorbar (maybe irregular intervals ?) could make it easier to identify these differences.*

We modified the color scale to make the differences easier to see.

ae) *With respect to the tables, would it be possible to also include the value of Delta-Delta? Maybe the authors could also give an indication of significance for DeltaA and DeltaO? I think this would make it even easier for the reader to get a feeling of the importance of Delta-Delta relative to DeltaA and DeltaO.*

We added these to the tables.

**3    Review 3:**

*1) The question that motivates the study is very interesting, but I think that the authors overstate their case too much to be convincing in their conclusions. My overall opinion is that this paper, although dealing with a novel and interesting question, is too modest in its present state. The simplicity of the numerical*

*design and the performed analyses suggest the authors overlook the complexity arising from coupling land, atmosphere and ocean in climate models, which is problematic for publishing in "Earth System Dynamics".*

*2) My main concern is that the reported differences between the interactive and fixed SST runs are weak and moderately significant, both at the large scale (in term of p-value in Table 1), and over the maps, in which the areas with insignificant changes are much larger than the ones with a significant change. The main exception is the SSTs themselves, but this is not very informative given that their variability is very different by construction in the two kinds of experiments (see also my comments a-b below).*

*Most of the recent papers that deal with tiny changes against the internal variability of the climate system use an ensemble approach to be more convincing from a statistical point of view, and I would like the authors explaining why they did not do the same.*

We analyzed equilibrium climate responses under fixed-SST and q-flux simulations with interannually constant forcing. Variability within this regime is expected to be independent of the detailed initial conditions, so running the simulations for a longer period plays the role of an ensemble in sampling the range of internal variability under each configuration (i.e. fixed-SST or q-flux, without or with irrigation).

*b) Even if we accept that the comparison of single members for each experiment is justified, information is missing in the paper regarding the experiment design and the subsequent statistical analysis:*

*a) I understood that the fixed SST simulations were analyzed over 50 years, while the SST forcing is available over 9 years only (1996-2004): how is it done, by cycling the 9-yr forcing over the 50 years? If so, it implies a very different variability to the one of the interactive SST simulations, for which we also need to know if you impose or not an increasing amount of GHG and aerosols, as this may induce a trend in addition to the seasonal to inter-annual variability (I'm not a specialist of slab oceans, so I need that kind of information to make sense of the results). I also wonder why the fixed SST runs do not rely more strongly on the AMIP protocol, which comes with a much longer SST forcing data set, starting in 1979.*

Different from the AMIP protocol, our goal was to examine equilibrium climate response to irrigation, with irrigation and other forcings held fixed at values from about the year 2000. The fixed SST runs applied an SST climatology derived by averaging the 1996-2004 values, so that SST stayed the same each year (not a 9-year cycle). Similarly, GHG and aerosol forcings were held at year-2000 values. We now clarify this in the paper.

*b) The above information is important since the significance of the analyzed differences is tested based on Student's t-test, which basically compares the mean difference to the variability (standard deviation) of the two compared samples. Regarding the test, I did not understand what was behind the following mention*

*(p3, L8) "with the degrees of freedom adjusted based on the lag-1 autocorrelation of the time series". Since the significance depends on the degrees of freedom, I recommend clarifying this point in the paper.*

Agreed, we now explain this better and give references.

*c) The paper relies on comparing the effect of irrigation in fixed and inter-active SST experiments. These effects are respectively called $\Delta_A$ and $\Delta_O$, and calculated as the difference between an irrigated and control experiment, in each of the confirmations. The rationale is that if there is a significant difference between $\Delta_A$ and $\Delta_O$, then it means that the interactive SST influences the response of the climate to irrigation. But we may imagine another explanation to a significant difference between $\Delta_A$ and $\Delta_O$, because the two control simulations must be different (AMIP and CMIP simulations are different), and may drive the differences between $\Delta_A$ and $\Delta_O$. Thus, I think the authors need to compare the differences between the two control experiments and the ones between $\Delta_A$ and $\Delta_O$ before concluding anything.*

Certainly the climate is not identical between the fixed-SST and q-flux configuration, although both are intended to simulate the contemporary climate state. To give a better sense of this, we now show the difference between the configurations in energy balance and in surface temperature. Our main goal here is to assess to what extent the modeled climate impact of irrigation changes based on which configuration is used, given that both have been previously employed for studies of irrigation impacts on climate but not systematically compared.

*3) I also regret that the analysis of the changes and the attempt to give them a physical explanation is rather superficial. The studied changes are induced by enhanced moisture input to the atmosphere over irrigated land, and the atmospheric humidity is not analyzed. The only circulation variable is the 300-mb height, and no mention is made to moisture convergence and convection for the atmospheric compartment, nor to monsoons and surface ocean currents. Yet, if there is an influence of the interactive ocean on the response of the climate to irrigation, it should imply that irrigation changes the ocean's behavior between the two interactive SST runs.*

We now show jet stream winds as an additional atmospheric variable. Changes in the monsoons are captured in the precipitation and pressure fields shown. Ocean currents are effectively held fixed with our q-flux model configuration, which we now mention as another possible influence mechanism of irrigation that is not captured in the model runs analyzed here.

*4) An illustrative example of the overstatement and lack of physical insight that can be found throughout the paper is the analysis of the precipitation changes over eastern Africa. We are asked to compare the $\Delta_A$ and $\Delta_O$ in precipitation in MAM over eastern Africa in Fig. 5, but there is almost nothing! I'm not even sure there would be something discernable with a magnifying glass. This*

*extremely weak change receives the longest explanation of the entire paper, with a 10-line paragraph, but it ends with a rather weak and speculative conclusion (p7, L13-15): "Thus, ocean-atmosphere interactions may importantly affect the magnitude and location of non-local irrigation impacts on climate, such as those potentially implicated in precipitation trends in eastern Africa."*

We removed this paragraph in order to focus on the more prominent differences seen.

*p2, L22: summed should probably be replaced by averaged*

Modified

*p3, L10 and p4, L24: do you analyze the rms or the standard deviation? The latter seems more informative, as it excludes the effect of differing means.*

RMS – this includes both change between the means of the differences $\Delta_A, \Delta_O$ (nonzero mean $\Delta\Delta$) and spatial fluctuations that average out to zero globally.

*P4, L14: when analyzing large scale means (land vs ocean), it's abusive to write that "interactive SST spreads the cooling": you need maps to draw this conclusion.*

We refer there to the vertical extent of temperature change (not its horizontal extent), with reference to the differences in 300-mb geopotential height, which responds to the vertical integral of temperature in the atmosphere, as opposed to single-level measures such as surface air temperature. We now try to state this more clearly.

*Caption of Table 2 should mention that the values correspond to the northern hemisphere only.*
*Caption of Table 3 should mention that the studied variability (by means of rms) is spatial (and not temporal, which could be worth an analysis too).*
*P7, 1st L15-16: please clarify "supporting the role of air-sea interaction in driving the divergence in surface air temperature and geopotential height irrigation responses between the fixed-SST and interactive-SST simulations."*
*P7, L17-18: please clarify what is really meant by "the same stationary wave pattern [. . .] is found [. . .], with shifted phase"*
*P7, L5 (according to the numbering in the pdf): please clarify "The role of interactive SST in non-local irrigation climate forcing"*

We clarified these in the appropriate places.

**4 Review 4:**

*Authors should investigate/comment of the effects on the oceans of taking all the water from the rivers. I.e. less water reaching the ocean and the effect on ocean temperatures.*

We do not use a dynamic ocean model in the presented work, so ocean salinity patterns as influenced by freshwater flux changes do not affect model climate. Also, ocean extent is prescribed, so sea level does not change because of water diversion or other causes. We now mention that these are potential additional feedbacks of irrigation on the Earth system that could be investigated using more sophisticated model configurations.

*The analysis is performed with a focus on spatial impacts, but I think the study would benefit if also some more attention would be given to the temporal effects. I.e. do the impacts change between the different years that were simulated and if so what are the main reasons?*

The simulations here used steady 'present-day' (year-2000) forcings, and our analysis focuses on the mean equilibrium response. Previous work has looked at how the impact of irrigation on climate might vary depending on, for example, greenhouse gas concentrations [5, 4].

*My final concern is that only one model with each of the configurations is being run and analysed, but little consideration is given to the uncertainties associated with the different initial conditions and model configurations. The authors should at least test/discuss if the effects observed from different initial states are larger than the effects one obtains from the presence/absence of irrigation and/or interactive SST.*

We focus in this paper on perturbations of the equilibrium climate due to irrigation, which are expected to be insensitive to the precise initial conditions. We state that conducting similar experiments with different models and configurations is necessary to better understand how robust the effects seen are.

*P1L19: please elaborate on the factors that prevent the deduction of remote impacts based on observations.*

We now elaborate on this, noting that the propagation mechanisms of remote impacts may not be easily observable and that trends in observations are often dominated by the effects of other climate forcings [6, 7, 8].

*P2L7: Please specify the degree of amplification, i.e. was the amplification significant?*

The papers we cite here state that certain impacts of LCLUC were more pronounced in models with an interactive ocean than with a fixed ocean, but

did not quantify the statistical significance of the changes between the two configurations ($\Delta\Delta$ in our nomenclature).

*P2 1st Paragraph: Were any major reservoirs considered before water was taken from the groundwater resources? Were the volumes of water and groundwater abstractions of the same magnitude as actual measured irrigation? In many countries where irrigation was applied, rice is one of the major crops grown. Have the effects of the cropping methods (e.g. rice paddies result in standing water and therefore into additional standing water and evaporation) been considered in the model? Please comment/elaborate on these points in the manuscript.*

We now explain the irrigation module more extensively and provide the water volume. Reservoirs are not represented in the model version that we used. We now clarify that the applied irrigation amount is "based on combining maps of irrigated areas and crop types with crop-specific evapotranspiration scale factors, with a special allowance for maintaining a constant flood depth in paddy rice areas".

*P3L1: the simulations were run for 60 years, however the data available for the SST and the seas ice were only computed for 1996-2004. Does this short time span cover the natural variability of these variables? Please elaborate.*

In the fixed-SST runs, the model SST and sea ice were intended to represent typical conditions for around the year 2000 and were therefore obtained by averaging years before and after 2000, but did not have interannual variability. We now clarify this.

*P3 Fig1: I suggest using a different colour scale (not starting with white), as it is difficult to distinguish if for an area no/ little irrigation has been applied. Additionally, please elaborate why certain areas have high amounts of irrigation applied, although it is the season of high precipitation (e.g. India during the Monsoon).*

The irrigation amounts are from the cited data set. We now explain that for paddy rice areas, the applied irrigation is supposed to maintain flooding by compensating for an assumed-constant infiltration rate, which might be the cause of irrigation being specified even during parts of the monsoon season. The color scale has been modified so that smaller irrigation amounts are more visible.

*P3L8: Please elaborate why a lag-1 adjustment was performed. What is the physical meaning behind this? Does the system not exhibit any long term memory?*

Yes, this adjustment is intended to estimate empirically the effective number of degrees of freedom of the time series. We now explain this better and give

references. Long-term memory could be accounted for by basing the adjustment on higher-order autocorrelations as well, but in practice large correlations in the examined climate variables generally do not persist over long (multi-year) periods in this type of run [3], so the lag-1 autocorrelation together with a red-noise or Markov model of the time series is expected to give a fair approximation of the effective number of degrees of freedom [9, 10].

*P3L12-15: Why only northern Hemisphere? I would think that the other 8% of irrigation also merits consideration.*

Although our maps show the distribution of effects globally, for the seasonality we focus on the NH since that is where, with over 90% of the global applied irrigation amount, we would expect to see greater local impacts.

*P7L5-10: I would suggest moving this to the discussion section.*

We removed this paragraph to focus more on the strongest impacts seen.

**References**

[1] Gavin A. Schmidt, Max Kelley, Larissa Nazarenko, Reto Ruedy, Gary L. Russell, Igor Aleinov, Mike Bauer, Susanne E. Bauer, Maharaj K. Bhat, Rainer Bleck, Vittorio Canuto, Yong-Hua Chen, Ye Cheng, Thomas L. Clune, Anthony Del Genio, Rosalinda de Fainchtein, Greg Faluvegi, James E. Hansen, Richard J. Healy, Nancy Y. Kiang, Dorothy Koch, Andy A. Lacis, Allegra N. LeGrande, Jean Lerner, Ken K. Lo, Elaine E. Matthews, Surabi Menon, Ron L. Miller, Valdar Oinas, Amidu O. Oloso, Jan P. Perlwitz, Michael J. Puma, William M. Putman, David Rind, Anastasia Romanou, Makiko Sato, Drew T. Shindell, Shan Sun, Rahman A. Syed, Nick Tausnev, Kostas Tsigaridis, Nadine Unger, Apostolos Voulgarakis, Mao-Sung Yao, and Jinlun Zhang. Configuration and assessment of the GISS ModelE2 contributions to the CMIP5 archive. *Journal of Advances in Modeling Earth Systems*, 6(1):141–184, 2014.

[2] Ron L. Miller, Gavin A. Schmidt, Larissa S. Nazarenko, Nick Tausnev, Susanne E. Bauer, Anthony D. DelGenio, Max Kelley, Ken K. Lo, Reto Ruedy, Drew T. Shindell, Igor Aleinov, Mike Bauer, Rainer Bleck, Vittorio Canuto, Yonghua Chen, Ye Cheng, Thomas L. Clune, Greg Faluvegi, James E. Hansen, Richard J. Healy, Nancy Y. Kiang, Dorothy Koch, Andy A. Lacis, Allegra N. LeGrande, Jean Lerner, Surabi Menon, Valdar Oinas, Carlos Pérez García-Pando, Jan P. Perlwitz, Michael J. Puma, David Rind, Anastasia Romanou, Gary L. Russell, Makiko Sato, Shan Sun, Kostas Tsigaridis, Nadine Unger, Apostolos Voulgarakis, Mao-Sung Yao, and Jinlun Zhang. CMIP5 historical simulations (1850-2012) with GISS

ModelE2. *Journal of Advances in Modeling Earth Systems*, 6(2):441–477, 2014.

[3] N. Y. Krakauer, B. I. Cook, and M. J. Puma. Contribution of soil moisture feedback to hydroclimatic variability. *Hydrology and Earth System Sciences*, 14:505–520, 2010.

[4] Benjamin I. Cook, Sonali P. Shukla, Michael J. Puma, and Larissa S. Nazarenko. Irrigation as an historical climate forcing. *Climate Dynamics*, 44(5-6):1715–1730, 2015.

[5] B. I. Cook, M. J. Puma, and N. Y. Krakauer. Irrigation induced surface cooling in the context of modern and increased greenhouse gas forcing. *Climate Dynamics*, 37(7-8):1587–1600, 2011.

[6] Min-Hui Lo, Chien-Ming Wu, Hsi-Yen Ma, and James S. Famiglietti. The response of coastal stratocumulus clouds to agricultural irrigation in California. *Journal of Geophysical Research*, 118(12):6044–6051, 2013.

[7] Ross E. Alter, Eun-Soon Im, and Elfatih A. B. Eltahir. Rainfall consistently enhanced around the Gezira Scheme in East Africa due to irrigation. *Nature Geoscience*, 8:763–767, 2015.

[8] Philipp de Vrese, Stefan Hagemann, and Martin Claussen. Asian irrigation, African rain: remote impacts of irrigation. *Geophysical Research Letters*, 43(8):3737–3745, 2016.

[9] Richard H. Jones. Estimating the variance of time averages. *Journal of Applied Meteorology and Climatology*, 14:159–163, 1975.

[10] Christopher S. Bretherton, Martin Widmann, Valentin P. Dymnikov, John M. Wallace, and Ileana Bladé. The effective number of spatial degrees of freedom of a time-varying field. *Journal of Climate*, 12(7):1990–2009, 1999.

---

## Author Response (AR1)

Please find our revised manuscript attached, along with a version with the differences from the Discussion manuscript highlighted.

*Some readers might wonder why a long model run is argued to be comparable with the statistics that could be drawn from an ensemble. This correspondence is legitimate under ergodic assumptions in a sufficiently representative long-term experiment allowing the invariants of motion to be thoroughly manifested.*
*The debate on whether the experimental outcome is sensitive to the initial conditions also merits some clarification to the reader. On one hand, the transient dynamics may naturally be sensitive to the initial conditions, particularly in unstable system configurations where uncertainties propagate rapidly. These do not pose a fundamental problem here since transient dynamics are not the object of the study. On the other hand, for a given set of parameters and fixed model structure, the asymptotic behaviour of a dynamical system will define a statistically invariant outcome, consistent with equilibrium statistical physics (e.g. shaped by attractors in dissipative systems). That is, while the transient dynamics are indeed sensitive to the initial conditions, in dissipative systems the asymptotic behaviour will exhibit similar statistical physics irrespective of the initial conditions - for given model parameters and structure.*

We agree that this asymptotic regime is where we conduct the analyses presented. We show the approach to this steady-state behavior in the new Figure 2 and attempt to clarify our intention better in the Methods section.

*Further elaboration on the physical context of the problem, the experiments and results would also be highly beneficial to the paper. In this sense, the manuscript would benefit from placing the kinematic lessons into dynamic context, i.e. complementing a motion-descriptive with physical considerations that help the reader better understand the dynamics at play.*
*On a more specific note (as raised by one of the reviewers), the reported shift in the ITCZ would merit some brief additional comment based on supporting arguments available in the literature.*

We added discussion of the ITCZ changes seen and their likely physical-dynamic explanation with references to related work and relevant previous experiments, as well as more details on the experimental set-up.

*Overall, there are questions raised by the reviewers that might be wondered by the broader readership. By openly addressing them in the manuscript as done in the peer-review process, the authors will quench potential controversy before it has the chance to ignite.*
*At this stage, the authors are then encouraged to proceed with their review efforts, paying special attention to the recommendations arising in the peer-review process.*

Yes, as detailed in our earlier response, we have made many changes in the paper to address the peer reviewer comments.

[revised manuscript text omitted]

---

## Referee Report (RR1)

**Review: Ocean-atmosphere interactions modulate irrigation's climate impacts**

**1 General**

In general, I think the authors did a very good job at integrating most of the reviewers' suggestions, strongly increasing the manuscript's quality. However, in my opinion there still exists a major issue with the conclusions the authors draw from their results.

In the last part of their discussion section (p. 8 l. 30 ff) the authors claim that simulations with fixed SSTs or regional models may miss some important effects related to irrigation. The way the section reads, this can very easily be understood as meaning that the respective simulations (fixed SSTs or regional models) also do a worse job at representing real world irrigation impacts. This the authors have not shown in their study, e.g. there is no comparison to observations and there is also no comparison to any results from regional simulations. Here, it is equally plausible that the ocean model introduces some erroneous effects that lead to an amplification of the irrigation impacts that in reality does not exist. Thus, I think this statement should be revised carefully. It should very clearly be acknowledged that (in the present study) there is no evidence that the amplifying effects of the ocean model actually result in a better representation of irrigation effects with respect to reality.

As I have stated before I see a danger that this notion establishes grounds on which results of future modelling studies can easily be rejected merely because the model did not include an interactive ocean component. Because this is an important issue, the authors should either ensure that the respective passage can not be misunderstood or provide evidence for an actual improvement as a result of having an ocean model.

As a minor issue, it would be very nice to have the numbers for the fraction of the surface being significantly affected by irrigation (p. 6 l. 50 ff) also for the land surface only.

---

## Editor Decision (ED1)

**ESD-2016-23: Editor Decision Letter**

August $11^{th}$, 2016

Dear authors,

Thank your very much for your comments and diligences in response to the reviewer reports.

The scientific questions addressed in the manuscript are relevant to the scope of Earth System Dynamics, and some significant efforts have been made by the authors in addressing such questions.

However, the manuscript would considerably benefit from further elaboration, clarification and revision, with particular attention to the reviewer concerns. In this regard, the diligences conducted in response to the reviewers are an encouraging step forward and should definitely proceed.

Following the fertile discussion stage, I would highlight a few details that merit further discussion: the experiment design, the associated uncertainty debate, and the physical interpretations:

Some readers might wonder why a long model run is argued to be comparable with the statistics that could be drawn from an ensemble. This correspondence is legitimate under ergodic assumptions in a sufficiently representative long-term experiment allowing the invariants of motion to be thoroughly manifested.

The debate on whether the experimental outcome is sensitive to the initial conditions also merits some clarification to the reader. On one hand, the transient dynamics may naturally be sensitive to the initial conditions, particularly in unstable system configurations where uncertainties propagate rapidly. These do not pose a fundamental problem here since transient dynamics are not the object of the study. On the other hand, for a given set of parameters and fixed model structure, the asymptotic behaviour of a dynamical system will define a statistically invariant outcome, consistent with equilibrium statistical physics (e.g. shaped by attractors in dissipative systems). That is, while the transient dynamics are indeed sensitive to the initial conditions, in dissipative systems the asymptotic behaviour will exhibit similar statistical physics irrespective of the initial conditions - for given model parameters and structure.

Further elaboration on the physical context of the problem, the experiments and results would also be highly beneficial to the paper. In this sense, the manuscript would benefit from placing the kinematic lessons into dynamic context, i.e. complementing a motion-descriptive with physical considerations that help the reader better understand the dynamics at play.

On a more specific note (as raised by one of the reviewers), the reported shift in the ITCZ would merit some brief additional comment based on supporting arguments available in the literature.

Overall, there are questions raised by the reviewers that might be wondered by the broader readership. By openly addressing them in the manuscript as done in the peer-review process, the authors will quench potential controversy before it has the chance to ignite.

At this stage, the authors are then encouraged to proceed with their review efforts, paying special attention to the recommendations arising in the peer-review process.

I will be looking forward to the revised manuscript.

With very best wishes,

Rui Perdigão
(ESD Editor)

---

## Author Response (AR2)

Please find our revised manuscript attached.

**1   Referee 1 (report 2)**

*The authors have addressed my comments, and I just have one minor comment in their revised manuscript regarding the jet stream definition.*
*Please clearly define the "jet stream" in the revised manuscript, for example: at which level?*

We now do so.

**2   Referee 2 (report 1)**

*In the last part of their discussion section (p. 8 l. 30 ff) the authors claim that simulations with fixed SSTs or regional models may miss some important effects related to irrigation. The way the section reads, this can very easily be understood as meaning that the respective simulations (fixed SSTs or regional models) also do a worse job at representing real world irrigation impacts. This the authors have not shown in their study, e.g. there is no comparison to observations and there is also no comparison to any results from regional simulations. Here, it is equally plausible that the ocean model introduces some erroneous effects that lead to an amplification of the irrigation impacts that in reality does not exist. Thus, I think this statement should be revised carefully. It should very clearly be acknowledged that (in the present study) there is no evidence that the amplifying effects of the ocean model actually result in a better representation of irrigation effects with respect to reality.*

We added to the Discussion "We note that the present work only shows that interactive SST alters the climate effects of irrigation in GCM simulations. We have not conducted comparisons with observations to demonstrate that simulations with interactive SST actually represent irrigation effects on climate better than simulations with fixed SST. Showing that the responses seen with interactive SST are consistent across different GCMs could increase confidence that the results reported here are physically meaningful and not an artifact of a particular model configuration."

**3   Referee 4 (report 3)**

*Overall, it is clear that the authors made an effort to integrate the reviewers? comments into the revised version of the paper.*

*However, I still think that the manuscript would benefit if the authors would treat the interactions between the modelled components from a more physical perspective and thoroughly discuss the processes behind the numerical experiments that lead to the differences presented (as already mentioned the first review).*

*This is of particular importance as the journal is interdisciplinary and therefore has a readership from various disciplines (which might be different to the authors? core expertise and not share the same background knowledge).*

*Especially the results section would benefit from a better physically based explanation, instead of a mere description.*

*E.g. P3 L41: "As expected, . . . ". Such statements are not helpful for the readership with various backgrounds. Please elaborate. . .*

We expanded the Introduction to provide a general physical explanation for the expectation that climate impacts of forcing such as irrigation would be more widespread in an interactive SST model configuration, compared to fixed SST, and refer to it in the Results section.

*In addition, Figures 3 to 8 show currently the absolute change (i.e. delta_A or delta_O).*

*However, for the readers with different backgrounds the absolute numbers of change have little meaning. For example, it might be difficult to judge for the reader if the plotted changes in the jet stream speed are substantial differences or if the difference and hence the improvement is only marginal. I therefore suggest adding plots (maybe into the supplements) that show not the absolute change but rather the change relative to the control case. This would allow the reader to understand the quantitatively the role of the ocean-atmosphere interactions.*

Average absolute values from the control simulation for each variable are given in Table 1, which together with Table 3 can be used to get a sense of the relative size of the typical delta_A, delta_O seen. We now also mention in the text the order of magnitude of the relative changes for variables that are likely to be less familiar to readers.

*Finally, I would like to see the authors discussing if their model results are actually depicting what can be observed in reality or if there is also the possibility that the results could be a model artefact.*

As mentioned in our reply to Referee 2, we now address this point more prominently to the Discussion.

*The authors should add a statement on the amounts of groundwater abstracted (i.e. if the amounts of water abstractions are actually possible to be extracted (quantitatively))*

Added now.

*P 1 Abstract L 3: 'irrigation climate impacts', please consider rephrasing*

On rereading, the sentence seems clear to us.

*P2: L8: '... show that the responses to these LCLUC forcing are amplified ...'' Can the authors please elaborate on the reasons for this.*

Added now.

*P3 Fig1: When printed on paper, it is difficult to distinguish the light yellow (little irrigation) from the white background (no irrigation). Maybe you can add grey color for no irrigation or begin the color ramp with a darker color.*

We now use gray to depict land areas with no irrigation.

*Figure 4: add explanation of grey areas to captions.*

Added.

*Figure 3-6: I suggest revert the color scale of the Figures to match the one of Figure 8 (with negative changes to be red and positive changes to be blue, as this is more intuitive to interpret for the reader).*

For temperature (Figures 3-4), red seems intuitive to denote heating and blue to denote cooling. For other variables such as jet stream velocity (Figures 5-6) where there is no particular color connotation for higher vs. lower values, a blue to red scale is more conventional, in our experience, than red to blue.

[revised manuscript text omitted]